# Activation Compression in LLMs: Theoretical Analysis and Efficient Algorithm

## Abstract

Training large language models (LLMs) is highly memory-intensive, as training must store not only weights and optimizer states but also intermediate activations for backpropagation. While existing memory-efficient methods largely focus on gradients and optimizer states, activation compression is less well established due to the lack of LLM-tailored theory and guarantees. In this work, we develop a theoretical framework showing that activation compression is safe for linear operators when activation compression is unbiased, but problematic for nonlinear ones. We further derive gradient variance bound and establish convergence guarantees for applying activation compression to all linear operators under the standard $L$-smoothness assumption, showing that it does not change the convergence rate. Guided by the theory, we propose an activation–gradient co-compression method that reuses low-rank activation factors to compress linear-layer gradients without extra computation or additional gradient error. We conduct extensive experiments on Qwen and LLaMA models using a pretraining benchmark and multiple fine-tuning benchmarks to validate our theory and demonstrate competitive performance of our method in both accuracy and compression efficiency. We provide our code in the supplementary material for reproducibility.

## 1. Introduction

As large language models (LLMs) continue to scale, they have demonstrated strong performance across a broad range of challenging tasks. However, this comes with significant memory overhead during training, since it requires storing the model parameters, gradients, optimizer states, and intermediate activations (Narayanan et al., 2021; Zhao

[1]Anonymous Institution, Anonymous City, Anonymous Region, Anonymous Country. Correspondence to: Anonymous Author <anon.email@domain.com>.

Preliminary work. Under review by the International Conference on Machine Learning (ICML). Do not distribute.

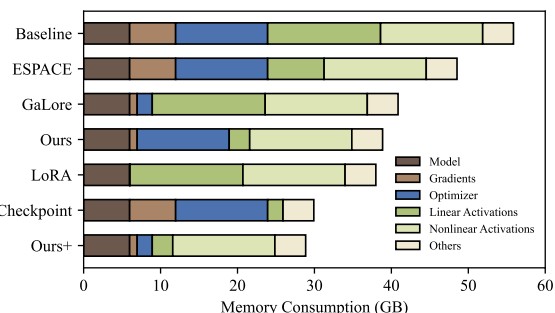

*Figure 1.* Estimated GPU memory consumption during fine-tuning of a LLaMA3-3B model on GSM8K dataset (batch size = 32).

et al., 2024; Wang et al., 2025b). As a result, memory consumption has become one of the primary bottlenecks in LLM training. Figure 1 illustrates a representative estimated GPU memory breakdown when fine-tuning LLaMA3-3B on dataset GSM8K with a batch size of 32.

To reduce the memory footprints of gradients and optimizer states, a number of memory-efficient training approaches have been proposed. Examples include parameter-efficient fine-tuning (Hu et al., 2022; Houlsby et al., 2019; Lester et al., 2021), gradient low-rank projection (Hao et al., 2024; Zhao et al., 2024) and quantization (Dettmers et al., 2022). Besides gradients and optimizer states, activation storage is also a major source of the memory overhead. To reduce activation memory, system-level techniques such as gradient checkpoint (Chen et al., 2016) and activation offloading (Rhu et al., 2016) use recomputation or I/O transfer, but often incur substantial overhead. Recently, several works (Sakr & Khailany, 2024; Wang et al., 2025b; Shamshoum et al., 2025) explore low-rank compression of the activation matrices in supervised fine-tuning (SFT) or LoRA settings, and further reduce the activation storage.

However, compressing activations can alter the backward computations along the entire gradient flow, potentially compromising optimization stability and model performance. Therefore, rigorous theoretical analysis and guarantees are essential for activation compression. Although prior theoretical works (Chen et al., 2021; Evans & Aamodt, 2021) establish convergence bounds on SGD with activation compression, they rely on assumptions that are difficult to sat-

isfy in multi-layer LLMs. Consequently, existing theories are still not well-developed for activation compression in LLM training, potentially exposing activation compression in LLMs to nontrivial risks. It may explain why activation compression has not attracted the same level of attention and adoption as PEFT or other memory-efficient training methods. This highlights a fundamental gap in the field: How to develop an LLM-tailored theoretical framework with practical guarantees for activation compression?

To bridge this gap, we propose a theoretical framework to analyze how activation compression influences training dynamics. We identify two critical questions at the operator[1] level that determine whether the adverse effects induced by compressing an operator's activation are safe: (i) Are the compression-induced gradient unbiased? and (ii) Will the gradient errors be propagated upstream to earlier operators and accumulated across layers?

To answer these questions, we prove that compressing the activation of a linear operator introduces neither (i) bias in the compression-induced gradient error nor (ii) upstream error propagation provided that the activation compression is unbiased. In contrast, activation compression for nonlinear operators induces both adverse effects, making the resulting degradation substantially severe. This analysis indicates that activation compression is safe for linear operators but not for nonlinear operators.

Furthermore, from a model-level perspective, we analyze mini-batch SGD when all linear operators employ activation compression, and derive a practical upper bound on the gradient variance. The bound decomposes into two additive parts: a mini-batch sampling variance term and a compression-induced variance term. Specifically, the compression term is a sum over operators, with each contribution governed by the second-order magnitude of the corresponding activation error. Together, these contributions enter additively, so the overall variance increase remains controlled. Building on this result, we establish convergence guarantees for training under the standard assumption that the objective is $L$-smooth. In particular, for the same number of gradient updates, the stationarity metric (measured by the expected norm of the full-parameter gradient) is at most a factor of $1 + \mathcal{O}\left(\left(\sum_l \mathbb{E}\|\Delta \boldsymbol{X}^l\|_F^2\right)^{\frac{1}{4}}\right)$ larger than that of uncompressed SGD, where $\sum_l \mathbb{E}\|\Delta \boldsymbol{X}^l\|_F^2$ aggregates the expected cumulative activation compression error across linear operators. Overall, these results suggest that compressing linear operators has only a mild impact on training dynamics and is practically safe.

Guided by theory, we further propose an efficient method that jointly compresses the activations and gradients for lin-

---

[1]Here, an *operator* corresponds to a **within-layer module** in Transformer and we will detail it in Section 3.1.

ear operators, without any additional computation overhead and gradient error beyond that already incurred by activation compression. The key insight is that once activation of an operator is low-rank, the corresponding weight gradient is naturally low-rank as well. Thus, leveraging the low-rank factors of the compressed activations, our method expresses and maintains the weight gradients of linear operators in a matching factorized low-rank form throughout training. As a result, it significantly reduces gradient memory on top of the savings from low-rank activation compression.

To validate our theory and evaluate the effectiveness of our method, we conduct extensive experiments on Qwen and LLaMA models across multiple pretraining datasets and fine-tuning datasets. Empirically, the results closely match our theory, and our method achieves competitive performance in both accuracy and compression efficiency compared to SOTA memory-efficient training approaches.

In summary, our contributions can be listed as follows:
(i) We develop a comprehensive theoretical framework for activation compression tailored to LLM training. (ii) We propose a simple and efficient method that jointly compresses activations and gradients for linear operators achieving substantial additional memory savings. (iii) We conduct extensive experiments to validate our theory and demonstrate the effectiveness of our method.

## 2. Related Work

**Memory Reduction for Gradients and Optimizer States.**
Many well-established methods, such as quantization and parameter-efficient fine-tuning (PEFT), have been proposed to reduce the memory footprint of gradients and optimizer states. Quantization reduces memory via low-bit representations, and most commonly for compressing model weights (Lee et al., 2024; Dettmers et al., 2023; Wang et al., 2025a), and optimizer states (Dettmers et al., 2022; Li et al., 2023). PEFT freezes most pretrained weights and optimizes only a small set of parameters. Examples include low-rank adaptation methods that parameterizes weight updates with low-rank factors (Hu et al., 2022; Zhang et al., 2023; Valipour et al., 2023) and additive approaches that introduce lightweight trainable modules or learnable prompts while keeping the backbone fixed (Houlsby et al., 2019; Lester et al., 2021; Li & Liang, 2021). However, the limited update capacity of these approaches can result in inferior performance compared with full SFT (Xia et al., 2024). To alleviate this limitation, Flora (Hao et al., 2024), Galore (Zhao et al., 2024), and subsequent variants (Muhamed et al., 2024; Xiao et al., 2025; Liang et al., 2024) propose projecting gradients onto a low-rank subspace for optimizer updates. Despite these advances, existing methods cannot alleviate the activation memory overhead, and their theoretical guarantees as well as empirical accuracy still leave room

for improvement. In this paper, we propose an efficient method that jointly compresses the activation and gradient of without any additional computation overhead beyond the cost of activation compression.

**Activation Memory Reduction Methods and Theory.** Backpropagation requires access to intermediate activations from the forward pass which must be stored in GPU memory. Since this footprint scales roughly linearly with context length and model size, activation storage also dominates training memory, making activation reduction crucial for PEFT. System-level techniques such as gradient checkpoint (Chen et al., 2016) and activation offloading (Rhu et al., 2016) can substantially reduce activation memory via recomputation or I/O transfers. However, typically they may incur significant computation or communication overhead. To mitigate this problem, a number of works propose compressing activations via quantization (Chakrabarti & Moseley, 2019), data encoding (Jain et al., 2018; Vu et al., 2024), or low-rank approximation (Nguyen et al., 2024; 2025) in CNNs, ResNets, and other neural architectures. Theoretical studies such as AC-GC (Evans & Aamodt, 2021) and ActNN (Chen et al., 2021) also establish convergence guarantees for SGD with activation compression. However, these studies typically rely on assumptions that the compression-induced gradient errors are unbiased and upper-bounded, which are difficult to be satisfied in multi-layer LLMs.

In the context of LLM training, ESPACE (Sakr & Khailany, 2024), CompAct (Shamshoum et al., 2025) and CoLA (Liu et al., 2025) study activation compression for pretraining or SFT, while HYC-LoRA (Wang et al., 2025b) and LoRAct (Shi et al., 2025) focus on the LoRA fine-tuning setting. CompAct projects activations to a low-dimensional space using Gaussian random projection and performs optimizer updates in the resulting low-rank subspace. However, the framework is restricted to low-rank activation compression via random projections, which limits its applicability, and it often underperforms SFT in terms of accuracy. CoLA modifies the model architecture to enable low-rank activation compression, making it efficient for pretraining but less suitable for fine-tuning. Besides, note that in practice works only apply activation compression to the linear operators. However, they lack principled theoretical guarantees and explaination of this design choice. To address the above gaps, this paper aims to develop a comprehensive theoretical framework for activation compression.

# 3. Theoretical Framework for Activation Compression in LLM Training

In this section, we develop a practical theoretical framework for activation compression tailored to LLM training. We show that it is safe for linear operators under unbiased compression but can be problematic for nonlinear operators. We also provide convergence guarantees for mini-batch SGD when compressing all linear activations.

## 3.1. Activation Compression Mechanism and Definitions

Figure 2a illustrates the operator-level workflow of activation compression in one training step. In our framework, an *operator* refers to a computational node in the training computation graph. For decoder-only Transformers, it typically corresponds to a **within-layer module**, such as a linear projection (e.g. Q/K/V projections in attention) or a nonlinear operation (e.g. RMSNorm). For clarity and rigor, we formalize the workflow and define the notations below.

**Definition 3.1.** During the forward pass, an operator $f$ (parameterized by $W$) computes its output $Z = f(X; W)$ exactly from the current input $X$. Meanwhile, the input activation $X$, which is needed for backward computation, is compressed and stored in a compact form, typically as a low-rank approximation $UV^\top$. When computing gradients in the backward pass, the activation should be reconstructed as $\hat{X} = UV^\top$, with the activation error defined as $\Delta X \triangleq \hat{X} - X$. The operator receives the incoming gradient $\nabla_Z \mathcal{L}$ of the loss $\mathcal{L}$ w.r.t. its output $Z$, and uses $\hat{X}$ and $\nabla_Z \mathcal{L}$ to compute (i) the gradient $\hat{G}_W$ w.r.t $W$ for parameter update, and (ii) the gradient $\hat{G}_X$ w.r.t. its input for propagation to upstream operators. For completeness, we denote by $G_X$ and $G_W$ the exact gradients with respect to $X$ and $W$ in the standard (uncompressed) backward computation (i.e., computed using $X$).

By the chain rule, $G_W = \nabla_Z \mathcal{L} \cdot \frac{\partial Z}{\partial W}\big|_{(W, X)}$ and $\hat{G}_W = \nabla_Z \mathcal{L} \cdot \frac{\partial Z}{\partial W}\big|_{(W, \hat{X})}$ where "·" denotes the appropriate tensor contraction. $G_X$ and $\hat{G}_X$ can be obtained analogously. Specifically, the linear operators are ubiquitous in transformers, with $Z = XW^\top$, $G_X = (\nabla_Z \mathcal{L})W$ and $G_W = (\nabla_Z \mathcal{L})^\top X$.

## 3.2. Linear vs. Nonlinear Operators: Bias and Error Propagation

Since an operator's activations are used in backpropagation to compute gradients, activation compression introduces gradient perturbations that may change training dynamics and destabilize optimization. Thus, a principled theoretical framework for training with activation compression is essential. From the classical convergence theory (Ghadimi & Lan, 2013) for SGD on $L$-smooth objectives, the convergence (in expectation) of SGD is determined by whether the stochastic gradients are unbiased, and the gradient variance magnitude determines the convergence rate. Thus, in this section, we consider the following two critical questions that determine whether compressing an operator's activations is acceptable: (i) Are the compression-induced gradients **unbiased**? and

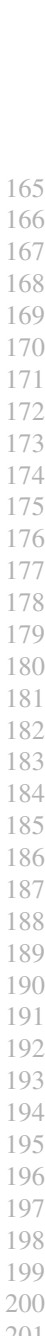

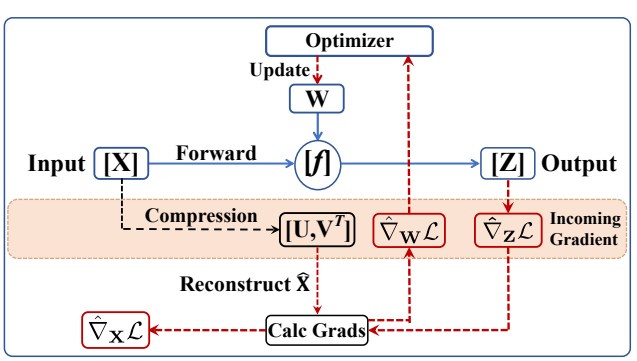

*(a)* Operator-level activation compression workflow.

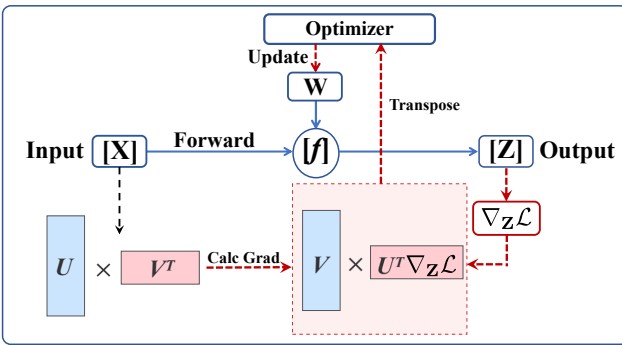

*(b)* Activation–Gradient Co-Compression Mechanism.

*Figure 2.* Comparison between the standard low-rank activation compression mechanism and our activation gradient co-compression method.

(ii) Will gradient errors **propagate upstream** to earlier operators and accumulate across layers, which may potentially lead to large gradient variance? Prior analyses (Evans & Aamodt, 2021; Chen et al., 2021) are developed mainly for conventional architectures such as CNNs, and often simply assume that compression-induced gradients are unbiased and uniformly upper-bounded. However, these assumptions may not hold in LLM training.

**I. Are compression-induced gradients unbiased?** In this section, we treat $\hat{X}$ as a stochastic estimator of $X$, and study the relationship between $G_W$ and $\mathbb{E}[\hat{G}_W]$ (and analogously between $G_X$ and $\mathbb{E}[\hat{G}_X]$). Based on Definition 3.1, the backpropagated parameter gradient is computed from the incoming gradient $\nabla_Z \mathcal{L}$ and the local Jacobian $\frac{\partial Z}{\partial W}$, which is a bivariate function of the input and parameter. When computing an operator's gradients, $\nabla_Z \mathcal{L}$ and $W$ are unchanged regardless of whether $X$ is compressed. Thus, $G_W$ and $\hat{G}_W$ can be interpreted as evaluations of the same mapping $\frac{\partial \mathcal{L}}{\partial W}$, differing only in the activation provided as input: $X$ for $G_W$ and $\hat{X}$ for $\hat{G}_W$.

Under this view, we consider the weight gradient as a mapping $\Phi$ of the activation. Let $\Phi(X) \triangleq \frac{\partial \mathcal{L}}{\partial W}(X)$ with fixed $W$ and given $\nabla_Z \mathcal{L}$. Take the Taylor expansion of $\hat{G}_W = \Phi(\hat{X})$ around $X$: $\Phi(\hat{X}) = \Phi(X) + J_\Phi(X)\Delta X + \frac{1}{2} H_\Phi(X)[\Delta X, \Delta X] + \mathcal{O}(\|\Delta X\|^3)$, where $\Phi(X) = G_W$, $J_\Phi(X)$ is the Jacobian, and $H_\Phi(X)[\cdot, \cdot]$ is the associated second-order bilinear form. When $f$ is a linear operator, if $\hat{X}$ is an unbiased estimator of $X$ (i.e., $\mathbb{E}[\Delta X] = 0$), then $\hat{G}_W$ is an unbiased estimator of $G_W$. This is because $f$ is linear in $X$ and so the Taylor expansion truncates after the first-order term. However, for a nonlinear operator $f$, $\hat{G}_W$ is generally biased, for $\frac{1}{2} H_\Phi(X)[\Delta X, \Delta X]$ depends quadratically on $\Delta X$ and typically has nonzero expectation. The analysis of relationship between $G_X$ and $\mathbb{E}[\hat{G}_X]$ is analogous and we let

$\Psi(X) \triangleq \frac{\partial \mathcal{L}}{\partial X}(X)$. Thus, we have the following Theorem.

**Theorem 3.2.** *Assume that* $\mathbb{E}[\Delta X] = 0$.
*(a) For linear operator* $f$: $\mathbb{E}[\hat{G}_W] = G_W, \mathbb{E}[\hat{G}_X] = G_X$.
*(b) For nonlinear operator* $f$:

$$\mathbb{E}[\Delta G_W] = \frac{1}{2}\,\mathbb{E}\big[H_\Phi(X)\big[\Delta X, \Delta X\big]\big] + \mathcal{O}\big(\mathbb{E}[\|\Delta X\|^3]\big),$$

$$\mathbb{E}[\Delta G_X] = \frac{1}{2}\,\mathbb{E}\big[H_\Psi(X)\big[\Delta X, \Delta X\big]\big] + \mathcal{O}\big(\mathbb{E}[\|\Delta X\|^3]\big),$$

*where* $\Delta G_W \triangleq \hat{G}_W - G_W$, *and* $\Delta G_X \triangleq \hat{G}_X - G_X$.

Theorem 3.2 shows that if the activation compression is unbiased, the compression-induced gradient estimators are also unbiased for linear operators; whereas for nonlinear operators they are generally biased, with the bias dominated by the second-order Taylor remainder. All proofs of the theorems are deferred to Section B.

**II. Will Errors Propagate Upstream?** The above analysis assumes that an operator's incoming gradient is exact, which may not always hold. In particular, an operator's gradient w.r.t. its input $G_X$ will become its preceding operators' incoming gradient during backpropagation. Thus, errors in $G_X$ can propagate backward and corrupt upstream gradient computations, and can be difficult to mitigate. Consequently, the gradient error $\hat{G}_X - G_X$ is a key factor governing the adverse effects of activation compression.

We have previously shown that for a linear operator $f$, the compressed gradient estimator is unbiased, namely, $\mathbb{E}[\hat{G}_X] = G_X$. In fact, we can further prove that the gradient w.r.t. its input is exact: $G_X = \hat{G}_X = \frac{\partial \mathcal{L}}{\partial Z} W$, implying that activation compression introduces no error in the input gradient. This holds because the operator's incoming gradient $\frac{\partial \mathcal{L}}{\partial Z}$ is identical with or without activation compression, and for a linear mapping $f$ the Jacobian $\frac{\partial Z}{\partial X}$ is independent of $X$. Conversely, we can also establish that if activation

compression introduces no error in the input gradient, then the operator must be linear. Below, we present a theorem that quantifies how compressing an operator's activations affects gradient computations in upstream operators.

**Theorem 3.3.** *Consider only compressing the activation of a single operator $f^l$ at depth $l$. For an upstream operator at depth $i < l$, let $\Delta \boldsymbol{G}_{\boldsymbol{X}}^i$ be the resulting gradient error w.r.t. the operator's input.*

*(a) $f^l$ is a linear operator $\iff \Delta \boldsymbol{G}_{\boldsymbol{X}}^l = \boldsymbol{0} \quad \forall \boldsymbol{X}, \hat{\boldsymbol{X}}, \frac{\partial \mathcal{L}}{\partial \boldsymbol{Z}}$.*

*(b) When $f^l$ is a non-linear operator:*

$$\Delta \boldsymbol{G}_{\boldsymbol{X}}^i = \Delta \boldsymbol{G}_{\boldsymbol{X}}^l \cdot \mathcal{J}^{l-1} \cdot \mathcal{J}^{l-2} \cdots \mathcal{J}^i,$$

*where $\mathcal{J}^i$ is the Jacobian of the $i$-th operator, and "$\cdot$" denotes tensor contraction.*

Theorem 3.3 shows that when the product of operator Jacobians exceeds 1, gradient errors can be amplified when compressing the activations of a nonlinear operator. On the other hand, compressing the activations of a linear operator does not affect upstream gradient computations.

Theorem 3.2 establishes that, given an exact incoming gradient, the gradient computed for a linear operator under activation compression is unbiased. Theorem 3.3 further shows that compressing the activations of a linear operator does not corrupt the incoming gradients of upstream operators. Taken together, the two theorems suggest that applying activation compression to all linear operators is safe, thereby addressing the two critical questions raised above. In contrast, applying activation compression to non-linear operators can destabilize training. Consistent with our findings, existing approaches (Shamshoum et al., 2025; Liu et al., 2025; Sakr & Khailany, 2024) predominantly compress activations in LLM's linear layers.

### 3.3. Gradient Variance Bounds and Convergence Guarantees

In this section, we initially derive upper bounds on the gradient variance under SGD with activation compression of **linear operators**, which is a key quantity directly governing the convergence speed. Following (Ghadimi & Lan, 2013; Chen et al., 2021), let $g$ be the gradient vector of the whole model, with variance $\mathrm{Var}(g) = \mathbb{E}\|g\|^2 - \|\mathbb{E}[g]\|^2$. Let $\Delta X^l$ be the activation error of the $l$-th linear operator, and $\boldsymbol{G}_{\boldsymbol{Z}}^l$ be the gradient w.r.t. $\boldsymbol{Z}$ at the $l$-th linear operator.

**Theorem 3.4.** *Assume that activation compression randomness is independent of mini-batch sampling. $\mathrm{Var}(g)$ is upper-bounded by:*

$$\frac{H(N-B)}{BN} + \frac{1}{B^2} \sum_l \mathbb{E}\big\|(\boldsymbol{G}_{\boldsymbol{Z}}^l)^\top \Delta \boldsymbol{X}^l\big\|_F^2,$$

*where $N$ is the total number of data samples, $B$ is the mini-batch size, and $H$ is a data-dependent constant induced by mini-batch sampling.*

As can be seen, the upper bound above consists of two terms. The first term is the variance induced by mini-batch sampling, and the second term is the variance induced by activation compression, whose magnitude depends on the sum of activation errors across operators. The bound indicates that contributions from mini-batch sampling and activation compression add linearly to the total gradient variance, implying that the total variance can be controlled.

Following the classical work on non-convex optimization (Ghadimi & Lan, 2013), for a fixed number of training steps $T$, let $\tau$ be sampled uniformly at random from $\{1, \ldots, T\}$, we measure convergence in terms of the expected gradient norm $\mathbb{E}\|\nabla f(\boldsymbol{x}_\tau)\|_2$. We define $\mathcal{U}_T^S$ as an upper bound on $\mathbb{E}\|\nabla f(\boldsymbol{x}_\tau)\|_2$ attained by standard SGD, and $\mathcal{U}_T^C$ as an upper bound on $\mathbb{E}\|\nabla f(\boldsymbol{x}_\tau)\|_2$ attained by compressed SGD. For standard SGD, $\mathcal{U}_T^S = \mathcal{O}(\frac{\sqrt{L}}{\sqrt{T}} + \frac{(\sigma^2 L)^{1/4}}{T^{1/4}})$, where $\sigma^2$ is variance upper bound of the gradient (Ghadimi & Lan, 2013). Similarly, we can establish convergence guarantees for SGD with activation compression: $\mathcal{U}_T^C = \mathcal{O}(\frac{\sqrt{L}}{\sqrt{T}} + \frac{(\sigma_c^2 L)^{1/4}}{T^{1/4}})$, where $\sigma_c^2$ is the gradient variance upper bound obtained in Theorem 3.4. To more intuitively illustrate how activation compression affects the convergence rate, we present the following Corollary.

**Corollary 3.5.** *When the loss is $L$-smooth with bounded gradient, $N \gg B$ and $T$ is sufficiently large, we have*

$$\frac{\mathcal{U}_T^C}{\mathcal{U}_T^S} \leq 1 + \mathcal{O}\left( (\sum_l \mathbb{E}\|\Delta \boldsymbol{X}^l\|_F^2)^{1/4} \right).$$

*where $\Delta \boldsymbol{X}^l$ is the activation error in the $l$-th linear operator introduced by activation compression.*

Theorem 3.5 shows that, for the same number of iterations $T$, the stationarity bound under activation compression is at most a $T$-independent multiplicative factor larger than that of standard SGD. This indicates that activation compression leads to a mild, controlled degradation in convergence.

## 4. Activation–Gradient Co-Compression

Existing approaches (Shamshoum et al., 2025; Liu et al., 2025; Sakr & Khailany, 2024) predominantly compress activations in the linear layers of LLMs during pretraining and SFT, most commonly via low-rank approximations. However, they either cannot reduce gradient memory overhead (Liu et al., 2025; Sakr & Khailany, 2024) or suffer from significant limitations in practice (Shamshoum et al., 2025). In this section, we propose a simple yet efficient method that jointly compresses activations and gradients for linear operators, achieving substantial additional memory savings

and greater generality. Since nearly all learnable parameters in Transformer reside in linear layers, our method can markedly reduce the overall memory footprint of gradients.

Our key insight is direct yet instructive: the weight gradient is given by $\boldsymbol{G_W} = (\frac{\partial \mathcal{L}}{\partial \boldsymbol{Z}})^\top \boldsymbol{X}$ for a linear operator, which implies that the rank of the gradient is at most the rank of the activation, i.e., $\mathrm{rank}(\boldsymbol{G_W}) \leq \mathrm{rank}(\boldsymbol{X})$. Therefore, once the activation is low-rank, the corresponding weight gradient is naturally low-rank as well. This motivates us to accordingly compress and store the gradients in a low-rank form, without introducing the additional error.

Suppose that the compressed activation $\hat{\boldsymbol{X}}_{m \times d}$ is stored in a factorized low-rank form $\boldsymbol{U}_{m \times k}$ and $\boldsymbol{V}_{d \times k}$, such that $\hat{\boldsymbol{X}} = \boldsymbol{U} \boldsymbol{V}^\top$, where $k$ is the rank. The corresponding weight gradient can then be written as

$$\hat{\boldsymbol{G}}_{\boldsymbol{W}} = (\frac{\partial \mathcal{L}}{\partial \boldsymbol{Z}})^\top (\boldsymbol{U} \boldsymbol{V}^\top) = \left( (\frac{\partial \mathcal{L}}{\partial \boldsymbol{Z}})^\top \boldsymbol{U} \right) \boldsymbol{V}^\top.$$

This shows that the gradient naturally admits a factorized representation as the product of two factors with rank $k$. As a result, during backpropagation we only need to compute and store the two factors rather than reconstructing the full gradient matrix. Figure 2b illustrates this workflow.

After the gradients for the entire model are computed, they are passed to the optimizer for states updates. Notably, given a low-rank gradient, the optimizer can update its states **in two strategies**: (i) reconstruct the full gradient and perform a standard update; or (ii) perform a GaLore-style update (Zhao et al., 2024), maintaining optimizer states in a compressed low-rank form for additional memory savings.

Therefore, compared with activation-only compression methods (Liu et al., 2025; Sakr & Khailany, 2024), our activation–gradient co-compression approach achieves substantial additional memory savings. Unlike CompAct (Shamshoum et al., 2025), which is limited to random projections, our framework supports general low-rank approximation techniques, providing greater generality and typically higher fidelity. Moreover, our method offers two desirable properties. First, gradient compression incurs no additional computational overhead, since we obtain a low-rank representation of the gradient matrix by leveraging the factorized activation, rather than explicitly decomposing the gradient. Second, gradient compression introduces no extra approximation error beyond that already incurred by activation compression.

## 5. Experiments and Analysis

In this section, we conduct extensive experiments and provide analyses of our empirical results. Section 5.1 first verifies our theoretical conclusions. Section 5.2 then evaluates the effectiveness of our proposed method and compare

it with other memory-efficient training methods.

**Experimental Setup** We conduct experiments primarily on fine-tuning, with additional pretraining results to evaluate our theory and method. For fine-tuning, we fine-tune LLaMA3-3B (Grattafiori et al., 2024) and Qwen3-4B (Yang et al., 2025) on GLUE (Wang et al., 2018), GSM8K (Cobbe et al., 2021), and MATH (Hendrycks et al., 2021). For pretraining, we follow prior work (Zhao et al., 2024; Liu et al., 2025) and use the C4 dataset (Raffel et al., 2020), training a LLaMA3-1B model. We evaluate our method with two compression ranks: 8 and 32. For activation compression, we adopt randomized singular value decomposition (RSVD) (Halko et al., 2011) and random projection (RP) (Shamshoum et al., 2025). In addition, we conduct ablation studies to validate the detailed results. We adopt the AdamW optimizer with an initial learning rate of $2 \times 10^{-5}$, and fix the random seed to $42$ for reproducibility. All fine-tuning experiments are conducted in bfloat16 (BF16) precision using PyTorch on a single NVIDIA A800-80GB GPU. We do not use gradient checkpointing for additional activation reduction. Due to space limitations, additional details and results are in Appendix A.

### 5.1. Empirical Verification of the Theory

To empirically validate our conclusions, We apply low-rank activation compression to different operators **separately** in the model and evaluate the resulting fine-tuning accuracy. A decoder-only Transformer generally consists of three nonlinear components: SiLU, RMSNorm, and Softmax. The Q/K/V projections in attention share the same activation, so they are compressed simultaneously; we denote this setting as Attn_In. In addition, we compress all linear operators simultaneously, which we denote as All_Linear. Table 1 reports the training top$-1$ accuracy across epochs for Qwen3-4B fine-tuned on GSM8K with rank 32 and batch size 8. We further present the training loss curves under activation compression for both the linear and nonlinear components in Figure 3, under the same experimental setting.

From Table 1, we observe that compressing activations of linear sublayers has wery little impact on training accuracy compared to the baseline (standard training without activation compression), with all variants reaching their peak performance at the third epoch. In contrast, compressing activations of nonlinear components leads to a significant degradation in performance. This effect is particularly pronounced for Softmax, where activation compression causes training to collapse. The observations are highly consistent with our theoretical results.

Figure 3 reports training loss curves under activation compression: SFT (left), all linear operators with ranks 8/32 (middle), and SiLU with ranks 8/32 as a representative non-

*Table 1.* Accuracy across epochs for different compressed layers. The best result in each row is highlighted in bold.

| Epoch
Layer | I | II | III | IV | V | VI | VII | VIII | IX | X |
|---|---|---|---|---|---|---|---|---|---|---|
| SFT | 0.786 | 0.780 | **0.788** | 0.773 | 0.757 | 0.739 | 0.735 | 0.726 | 0.722 | 0.716 |
| Attn_In | 0.778 | 0.784 | **0.785** | 0.765 | 0.757 | 0.740 | 0.728 | 0.729 | 0.731 | 0.726 |
| Attn_Out | 0.779 | 0.786 | **0.801** | 0.773 | 0.748 | 0.752 | 0.732 | 0.730 | 0.723 | 0.704 |
| MLP_In | 0.782 | 0.785 | **0.788** | 0.770 | 0.770 | 0.767 | 0.743 | 0.734 | 0.738 | 0.747 |
| MLP_Out | 0.784 | 0.779 | **0.796** | 0.776 | 0.761 | 0.752 | 0.742 | 0.732 | 0.727 | 0.732 |
| SiLU | **0.589** | 0.404 | 0.071 | 0.078 | 0.071 | 0.085 | 0.074 | 0.071 | 0.084 | 0.080 |
| RMSNorm | **0.650** | 0.550 | 0.490 | 0.459 | 0.454 | 0.433 | 0.428 | 0.416 | 0.423 | 0.428 |
| Softmax | 0.004 | **0.006** | 0.002 | 0.004 | 0.002 | 0.001 | 0.001 | 0.002 | 0.003 | 0.003 |
| All_Linear | 0.776 | 0.780 | **0.796** | 0.777 | 0.779 | 0.779 | 0.769 | 0.761 | 0.758 | 0.757 |

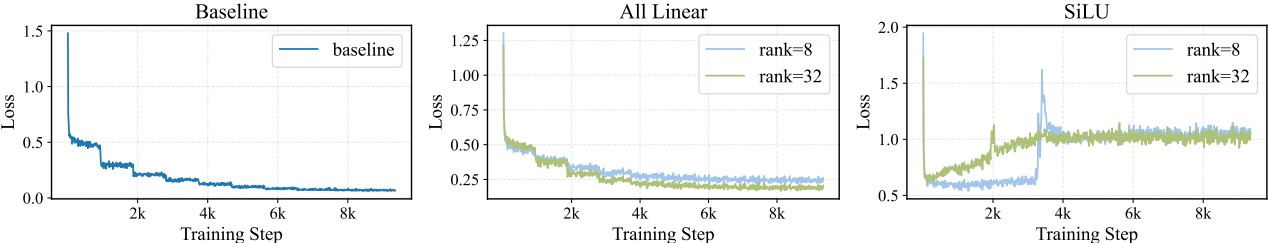

*Figure 3.* Comparison of training loss under different compression components.

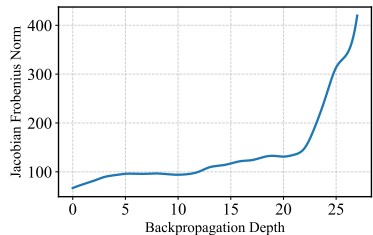

*Figure 4.* Frobenius norm of the Jacobian product.

linear operator(right). The figure shows that compressing activations of nonlinear components makes the loss difficult to decrease and prevents stable convergence. In contrast, compressing activations of linear operators does not impair training convergence, but only introduces minor perturbations to the optimization trajectory. This indicates that activation compression introduces only a small amount of additional gradient variance when applied to linear components, which supports our theoretical analysis.

Finally, we perform a layer-wise analysis by computing the Frobenius norm of the Jacobian product, providing empirical support for Theorem 3.3. We conduct the analysis at the granularity of individual layers. Since the Jacobian of layer is a large fourth-order tensor and infeasible to collect, we instead consider token-wise Jacobians. Figure 4 shows that the Frobenius norm of the Jacobian product grows with layer depth and significantly larger than 1, indicating that

error propagation can substantially amplify errors.

## 5.2. Method Evaluation and Comparison

In this subsection, we evaluate our method on multiple datasets and comparing its performance against other approaches. We evaluate two strategies of our method in Section 4, referred to as *Ours* and *Ours+*, respectively. *Ours* achieves better accuracy than *Ours+*, but it cannot reduce the memory footprint of the optimizer states. The selected competitors include LoRA, GaLore, and CompAct, which are representative memory-efficient training methods.

We conduct a comprehensive comparison from three perspectives: task accuracy, compression effectiveness, and training time. We use the estimated GPU memory breakdown and peak memory to assess compression effectiveness because the memory usage of individual components is difficult to measure directly.

**Accuracy Comparison.** Table 2 reports the accuracy of different methods with rank 8 on two base models, LLaMA3-3B and Qwen3-4B, evaluated on three benchmarks. On LLaMA3-3B, our method achieves competitive performance compared to the other efficient training approaches. On Qwen3-4B, all methods exhibit strong performance and our method also remains competitive and maintain comparable to SFT. The results demonstrate that our method preserves task accuracy.

*Table 2.* Accuracy under different methods on LLama3-3B and Qwen3-4B across GSM8K, Math, and GLUE benchmarks; the best result is highlighted in bold, and the second-best is underlined, excluding the SFT.

| Methods | LLama3-3B | | | Qwen3-4B | | |
|---------|-----------|-----------|-----------|-----------|-----------|-----------|
| | GSM8K | MATH | GLUE | GSM8K | MATH | GLUE |
| SFT | 0.466 | 0.124 | 0.840 | 0.788 | 0.499 | 0.845 |
| Ours | **0.417** | 0.111 | 0.771 | 0.796 | 0.493 | 0.823 |
| Ours+ | 0.398 | **0.117** | 0.800 | 0.780 | 0.489 | **0.824** |
| LoRA | 0.387 | 0.069 | **0.805** | 0.781 | **0.498** | **0.824** |
| CompAct | 0.290 | 0.082 | 0.734 | **0.798** | 0.482 | 0.776 |
| Galore | 0.367 | 0.108 | 0.727 | 0.791 | 0.491 | 0.773 |

*Table 3.* Training time of various methods.

| Methods | SFT | Ours+ | CompAct | Ours | LoRA | Galore |
|---------|-----|-------|---------|------|------|--------|
| Time (min) | 56.75 | 47.76 | 48.57 | 62.88 | 65.10 | 109.56 |

**Compression Effectiveness.** Figure 1 presents an estimated breakdown of GPU memory consumption for various memory-efficient methods with rank 32, illustrating their memory reduction effects. We use PyTorch's `saved_tensors_hooks` to estimate the breakdown. The experiments are conducted on LLaMA3-3B fine-tuned on GSM8K with a batch size of 32, under the FlashAttention setting (Dao et al., 2022). From the figure, we observe that *ours* substantially reduces the memory footprint of gradients and linear activations, and *Ours+* further achieves significant savings in optimizer states. Compared to the other methods, our approaches demonstrate a clear advantage in overall memory efficiency. In addition, under this experimental setting, we estimate the GPU memory breakdown of *Ours+* under different batch sizes in Figure 5, and report the corresponding breakdown (in GB) for batch size 32 in Table 4. We observe that as the batch size increases, activation storage becomes increasingly dominant.

**Training Time.** Table 3 reports the fine-tuning time (in minutes) of different methods on LLaMA3-3B for GSM8K. We observe that *Ours+* and CompAct are faster than the SFT, as they compress the optimizer-state matrices and thereby reduce the cost of parameter updates. *Ours* incurs additional

*Table 4.* GPU memory breakdown (GB) for SFT and Ours+ (batch size = 32).

| Part | SFT | Ours+ |
|------|-----|-------|
| Model | 5.980 | 5.980 |
| Nonlinear Activations | 13.275 | 13.275 |
| Linear Activations | 14.676 | 2.834 |
| Optimizer | 11.960 | 1.553 |
| Gradients | 5.980 | 0.776 |
| Others | 2.000 | 2.000 |

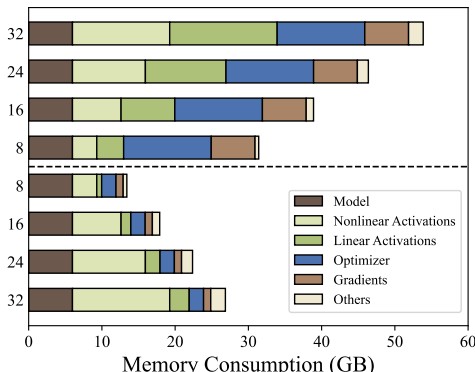

*Figure 5.* Estimated GPU memory usage under different batch sizes (y-axis: batch size).

computation mainly due to the low-rank factorization of activations, leading to an approximate $10.8\%$ increase in training time relative to the SFT. The results suggest that our methods achieve favorable time overhead.

## 6. Conclusion

In this paper, we develop an LLM-tailored theoretical framework to characterize how activation compression affects training dynamics. At the operator level, we show that activation compression is safe for linear operators but unsafe for nonlinear ones. We further derive gradient-variance bounds and establish convergence guarantees under standard $L$-smoothness assumptions when compressing activation of all linear opeartors. Our framework provides practical guidance and solid support for activation-compression paradigm. Guided by theory, we further propose an activation–gradient co-compression method that achieves substantial additional memory savings and broader applicability compared to existing approaches. Our method introduces no additional computational overhead and gradient error beyond that already introduced by activation compression. Extensive experiments validate our theory and show that our method remains competitive across all evaluation metrics.

## Impact Statement

This paper presents work whose goal is to advance the field of Machine Learning. There are many potential societal consequences of our work, none which we feel must be specifically highlighted here.

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

# A. Experimental Details and Additional Results

## A.1. Ablation Study

Here, we provide a fine-grained analysis of activation compression at the operator level. Specifically, we examine four operator types: Linear projection, RMSNorm, SiLU, and Softmax. We randomly sample 100 activation matrices for each type throughout the entire training process when fine-tuning LLaMA3-3B on GSM8K. Table 5 reports the detailed results. We evaluate two low-rank compression methods, RSVD and RP, and quantify the Frobenius-norm error between the original and compressed activations $\|\Delta X\|_F$ with rank 8 and 32. The experimental results indicate that operator activations are not strictly low-rank, which **supports our theoretical explanation** for why compressing the activations of linear operators has only a minor impact on model training. In contrast, prior work typically attributes the negligible accuracy degradation to the assumption that activations have very low rank—an explanation that is not sufficiently accurate.

In addition, by comparing $\|\Delta X\|_F$ of RSVD and RP, we observe that using RP to compress activations can be risky, as it may lead to an excessively large $\|\Delta X\|_F$. We also report in Tables 6 and 7 and Figure 6 a comparison between RSVD and RP under *ours* method (i.e., without optimizer compression) in terms of accuracy and loss convergence. Overall, RSVD outperforms RP, highlighting the necessity of our proposed co-compression framework, which is applicable to all low-rank compression algortihm.

*Table 5.* Frobenius-norm reconstruction error under different compression methods and ranks for selected layers on LLaMA3-3B evaluated with GSM8K.

| Layer | RSVD ($\|\Delta X\|_F$) | | RP ($\|\Delta X\|_F$) | |
|:---:|:---:|:---:|:---:|:---:|
| | rank=8 | rank=32 | rank=8 | rank=32 |
| Linear | 745.965 | 625.418 | 22398.720 | 11178.960 |
| RMSNorm | 966.600 | 812.140 | 28327.680 | 13894.080 |
| SiLU | 1508.120 | 1233.960 | 99136.000 | 52348.160 |
| Softmax | 51.116 | 44.729 | 1983.520 | 961.320 |

*Table 6.* Accuracy of LLaMA3 fine-tuned with RSVD and RP methods on the GSM8K dataset across different training epochs.

| Method \ Epoch | I | II | III | IV | V |
|:---:|:---:|:---:|:---:|:---:|:---:|
| RSVD | 0.353 | 0.388 | 0.409 | 0.412 | **0.413** |
| RP | 0.320 | 0.353 | 0.347 | **0.371** | 0.371 |

*Table 7.* Accuracy of LLaMA3 fine-tuned with RSVD and RP methods on the MATH dataset across different training epochs.

| Method \ Epoch | I | II | III | IV | V | VI | VII | VIII | IX | X |
|:---:|:---:|:---:|:---:|:---:|:---:|:---:|:---:|:---:|:---:|:---:|
| RSVD | 0.082 | **0.121** | 0.110 | 0.110 | 0.109 | 0.113 | 0.106 | 0.105 | 0.105 | 0.101 |
| RP | 0.019 | 0.036 | 0.054 | 0.042 | 0.043 | 0.054 | 0.047 | **0.055** | 0.055 | 0.052 |

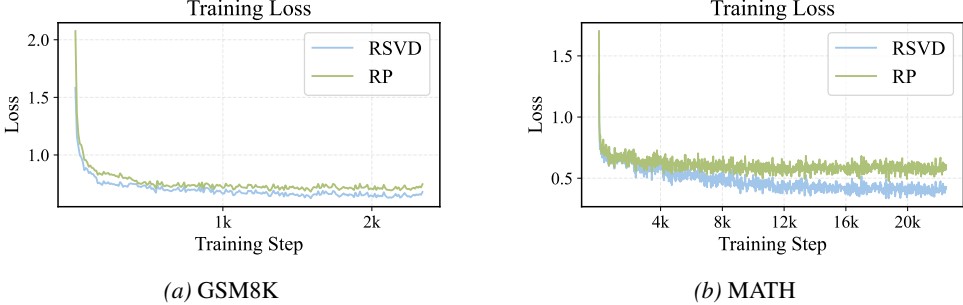

*(a)* GSM8K           *(b)* MATH

*Figure 6.* Training loss curves of RSVD and RP across different datasets.

### A.2. Peak Memory

Peak Memory refers to the maximum GPU memory footprint observed during training, and it is also an important metric for evaluating the effectiveness of memory-efficient methods in reducing GPU memory consumption. Table 8 reports the peak GPU memory usage of the baseline, *Ours*, and *Ours+* under the LLaMA GSM8K setting with The results indicate that our method can substantially reduce peak GPU memory usage and achieve strong compression efficiency.

**Implemental Details.** We measure peak GPU memory as the maximum CUDA memory allocated during training, obtained directly via PyTorch's `torch.cuda.max_memory_allocated()`. In addition, in our main experiments we provide an estimated breakdown of GPU memory usage. Since the memory consumption of individual components is difficult to measure directly, we use PyTorch's `saved_tensors_hooks` to record the sizes of activations produced by each operator before they are saved into the autograd `ctx` for backward computation. The measured activation memory closely matches the theoretically expected values.

### A.3. Pretraining Results

We also evaluate our method on a pre-training task, comparing its performance against GaLore (Zhao et al., 2024) and CoLA (Liu et al., 2025). Figure 7 presents the pre-training perplexity on the C4 dataset for the LLaMA3-1B model. Lower perplexity indicates better language modeling performance. We observe that our method outperforms CoLA during the first 3,000 training steps, but is later surpassed by CoLA. Galore performs strongly in the pre-training setting, although it shows relatively modest convergence and accuracy on fine-tuning tasks. However, in pre-training, the GPU memory footprint is dominated by activations. Since GaLore does not compress activations and instead only reduces the optimizer states, it has a major limitation.

*Table 8.* Comparison of Peak GPU Memory Usage

| Method | SFT | Ours | Ours+ |
|---|---|---|---|
| Peak GPU Memory (GB) | 65.282 | 53.117 | 42.729 |

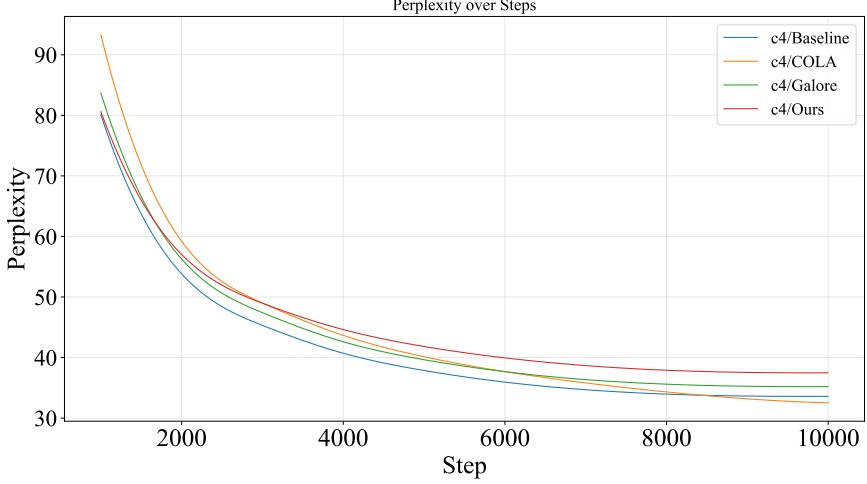

*Figure 7.* Pre-training perplexity on the C4 dataset for different methods

### A.4. Supplementary Results

In the main text, we provide training loss curves under activation compression for all linear layers and activations. Here, we further present the training loss curves under activation compression for two additional nonlinear operators, RMSNorm and Softmax. We also report the training loss curve of *Ours*.

Table 9 reports the training top−1 accuracy across epochs for Qwen3-4B fine-tuned on Math with rank 32 and batchsize 4 when compressing different operators. This serves as a supplementary experiment to Table 1. Figure 9 reports the Frobenius norms of more randomly sampled Jacobian matrices, serving as a supplement to Figure 4. This provides more convincing and broader evidence that the Frobenius norm of the Jacobian product grows with depth and becomes significantly larger than 1. Table 10 reports the estimated GPU memory breakdown of *Ours+* across all batch sizes for LLaMA3-3B fine-tuned on GSM8K, serving as a supplement to Table 4.

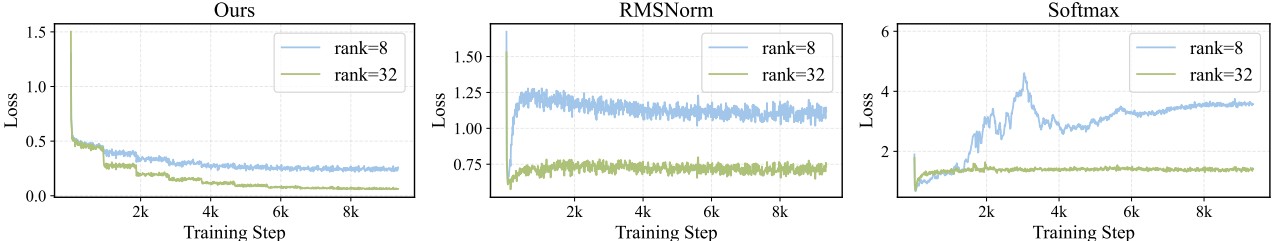

*Figure 8.* Comparison of Training Loss under Different Compression Components

*Table 9.* Accuracy across epochs for different compressed layers.

| Epoch
Layer | I | II | III | IV | V | VI | VII | VIII | IX | X |
|---|---|---|---|---|---|---|---|---|---|---|
| SFT | 0.480 | **0.491** | 0.477 | 0.493 | 0.477 | 0.463 | 0.470 | 0.471 | 0.473 | 0.471 |
| Attn_In | 0.500 | 0.501 | 0.490 | **0.502** | 0.477 | 0.453 | 0.438 | 0.433 | 0.434 | 0.444 |
| Attn_Out | 0.493 | **0.494** | 0.478 | 0.480 | 0.452 | 0.443 | 0.435 | 0.436 | 0.429 | 0.424 |
| MLP_In | **0.492** | 0.492 | 0.485 | 0.484 | 0.462 | 0.460 | 0.450 | 0.454 | 0.442 | 0.448 |
| MLP_Out | **0.494** | 0.492 | 0.489 | 0.483 | 0.460 | 0.454 | 0.438 | 0.431 | 0.432 | 0.431 |
| All_Linear | 0.480 | **0.491** | 0.477 | 0.493 | 0.477 | 0.463 | 0.470 | 0.471 | 0.473 | 0.471 |
| SiLU | **0.069** | 0.046 | 0.044 | 0.047 | 0.038 | 0.036 | 0.043 | 0.045 | 0.044 | 0.043 |
| RMSNorm | 0.418 | **0.456** | 0.450 | 0.430 | 0.410 | 0.412 | 0.409 | 0.398 | 0.399 | 0.402 |
| Softmax | 0.002 | 0.001 | 0.002 | 0.001 | **0.003** | 0.003 | 0.003 | 0.003 | 0.002 | 0.003 |

*Table 10.* GPU memory breakdown (GB) across batch sizes for SFT and Ours+.

| Part | SFT | | | | Ours+ | | | |
|---|---|---|---|---|---|---|---|---|
| | 8 | 16 | 24 | 32 | 8 | 16 | 24 | 32 |
| Model | 5.980 | 5.980 | 5.980 | 5.980 | 5.980 | 5.980 | 5.980 | 5.980 |
| Nonlinear Activations | 3.319 | 6.637 | 9.956 | 13.275 | 3.319 | 6.637 | 9.956 | 13.275 |
| Linear Activations | 3.669 | 7.338 | 11.007 | 14.676 | 0.734 | 1.434 | 2.134 | 2.834 |
| Optimizer | 11.960 | 11.960 | 11.960 | 11.960 | 1.553 | 1.553 | 1.553 | 1.553 |
| Gradients | 5.980 | 5.980 | 5.980 | 5.980 | 0.776 | 0.776 | 0.776 | 0.776 |
| Others | 0.500 | 1.000 | 1.500 | 2.000 | 0.500 | 1.000 | 1.500 | 2.000 |

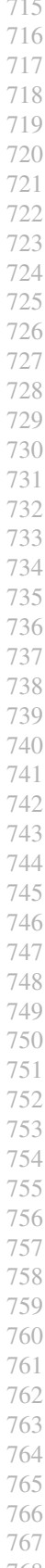

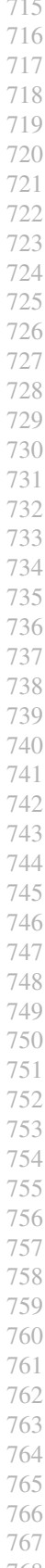

*(a)* Sample 1

*(b)* Sample 2

*(c)* Sample 3

*(d)* Sample 4

*(e)* Sample 5

*(f)* Sample 6

*(g)* Sample 7

*(h)* Sample 8

*(i)* Sample 9

*Figure 9.* Frobenius norm of randomly sampled Jacobian matrices.

## B. Proof

### B.1. Proof of Theorem 3.2

**(a) Linear operators.** When the forward operator $f$ is linear in $X$, gradient mappings $\Phi(X)$ and $\Psi(X)$ are linear functions of $X$.

For the weight gradient,

$$\hat{G}_W = \Phi(X + \Delta X) = \Phi(X) + \Phi(\Delta X). \tag{1}$$

Taking expectation and using $\mathbb{E}[\Delta X] = \mathbf{0}$ yields $\mathbb{E}[\hat{G}_W] = G_W$.

For the input gradient, the backward mapping is independent of the stored activation for linear operators, and thus $\mathbb{E}[\hat{G}_X] = G_X$.

**(b) Nonlinear operators.** When $f$ is nonlinear in $X$, the gradient mappings $\Phi$ and $\Psi$ are generally nonlinear.

Applying a second-order Taylor expansion of $\Phi(X + \Delta X)$ around $X$ gives

$$\Phi(X + \Delta X) = \Phi(X) + J_\Phi(X)[\Delta X] + \frac{1}{2}H_\Phi(X)[\Delta X, \Delta X] + \mathcal{O}(\|\Delta X\|^3). \tag{2}$$

where $D_\Phi(X)[\Delta X]$ is linear in $\Delta X$. Taking expectation and using $\mathbb{E}[\Delta X] = \mathbf{0}$ yields

$$\mathbb{E}[\Delta G_W] = \frac{1}{2}\mathbb{E}[H_\Phi(X)[\Delta X, \Delta X]] + \mathcal{O}\left(\mathbb{E}[\|\Delta X\|^3]\right). \tag{3}$$

The same argument applies to $\Psi$, yielding the stated result for $\mathbb{E}[\Delta G_X]$.

### B.2. Proof of Theorem 3.3

**(a) Linear operators.** In section a, we always discuss within a single layer, so we omit the superscript $l$.

($\Rightarrow$)We first specialize to the linear operator used in Transformers. Consider $Z = f(X; W) = XW^\top$, and let the upstream gradient be $G_Z \triangleq \frac{\partial \mathcal{L}}{\partial Z}$. By the chain rule (and as stated in Definition 3.1), the exact input gradient is

$$G_X = G_Z W. \tag{4}$$

Under activation compression, $X$ is stored in compressed form during the forward pass and reconstructed as $\hat{X}$ during backpropagation. The approximate gradients are computed by replacing $X$ with $\hat{X}$ in the backward computation. However, the expression in (4) does *not* depend on $X$. Therefore, we can remove the expectation of $\hat{G}_X$ based on theorem 3.2 and obtain a more robust conclusion.

$$\hat{G}_X = G_Z W = G_X, \tag{5}$$

which implies $\Delta G_X \triangleq \hat{G}_X - G_X = \mathbf{0}$ for all $X$ and $\hat{X}$. holds *deterministically*.

**Bias term.** The same "exactness" property also holds for the bias gradient if we consider the affine linear operator

$$Z = XW^\top + B. \tag{6}$$

then we have

$$G_B \triangleq \frac{\partial \mathcal{L}}{\partial B} = \frac{\partial \mathcal{L}}{\partial Z} = G_Z, \tag{7}$$

which again is independent of $X$, hence $\hat{G}_B = G_B$ for all $X, \hat{X}$.

($\Leftarrow$) We now prove the converse under the strengthened quantification :

$$\Delta G_X = \mathbf{0} \quad \forall X, \hat{X}, \forall G_Z \implies f \text{ is affine linear.} \tag{8}$$

By the chain rule, for a general operator $Z = f(X; W)$ we can write

$$G_X = G_Z \cdot \frac{\partial Z}{\partial X}\Big|_X, \qquad \hat{G}_X = G_Z \cdot \frac{\partial Z}{\partial X}\Big|_{\hat{X}}, \tag{9}$$

where "·" denotes the appropriate tensor contraction, consistent with Definition 3.1. Assumption (8) implies that for all $X, \hat{X}$ and all $G_Z$,

$$G_Z \cdot \left.\frac{\partial Z}{\partial X}\right|_X = G_Z \cdot \left.\frac{\partial Z}{\partial X}\right|_{\hat{X}}. \tag{10}$$

Since (10) holds for *all* upstream gradients $G_Z$, it follows that the multilinear map $\left.\frac{\partial Z}{\partial X}\right|_X$ must be identical for all $X$, i.e.,

$$\left.\frac{\partial Z}{\partial X}\right|_X = \left.\frac{\partial Z}{\partial X}\right|_{\hat{X}} \quad \forall\, X, \hat{X}. \tag{11}$$

Thus, $\frac{\partial Z}{\partial X}$ is independent of $X$ and equals a constant linear map, denoted by $A$. Consequently, $f$ has constant derivative and hence is an affine linear mapping.

**(b) unlinear operators.**  Reviewing Jacobian's definition $\mathcal{J}^i \triangleq \frac{\partial X^{i+1}}{\partial X^i}$, we apply the chain rule and observe that

$$G_X^i = G_X^{i+1} \cdot \mathcal{J}^i, \qquad i = 1, \ldots, l-1. \tag{12}$$

Briefly reviewing our method of storing activation values. We noticed that, for all operators $i < l$, the same local Jacobian $\mathcal{J}^i$ is used in both exact and compressed backpropagation. Therefore, the gradient estimation values satisfy

$$\hat{G}_X^i = \hat{G}_X^{i+1} \cdot \mathcal{J}^i, \qquad i = 1, \ldots, l-1. \tag{13}$$

Subtracting the two equations yields the error recursion

$$\Delta G_X^i = \hat{G}_X^i - G_X^i = (\hat{G}_X^{i+1} - G_X^{i+1}) \cdot \mathcal{J}^i = \Delta G_X^{i+1} \cdot \mathcal{J}^i, \tag{14}$$

Follow this equation, we unrolling this recursion from $l-1$ down to $i$

$$\Delta G_X^i = \Delta G_X^l \cdot \mathcal{J}^{l-1} \cdot \mathcal{J}^{l-2} \cdots \mathcal{J}^i, \tag{15}$$

## B.3. Proof of Theorem 3.4

**Step 1: Additivity of sampling variance and compression variance.**  Let g denote the vectorized parameter gradient produced by a mini-batch of size B. We decompose g as

$$g = \bar{g} + \Delta g, \qquad \bar{g} \triangleq \frac{1}{B} \sum_{b=1}^{B} g^{(b)}, \qquad \Delta g \triangleq \frac{1}{B} \sum_{b=1}^{B} \Delta g^{(b)}, \tag{16}$$

where $g^{(b)}$ is the (uncompressed) per-sample parameter gradient determined by the sampled data, and $\Delta g^{(b)}$ is the perturbation induced by using reconstructed activations in backpropagation.

Considering that g is the concatenation of all the matrices in the model after being vectorized, since we have previously proved that only the linear layers need to be compressed, i.e., $\mathbb{E}[\Delta G_W] = \mathbf{0}$ for all linear layers, we have $\mathbb{E}[\Delta g] = \mathbf{0}$ .Because our compression strategy and sampling $\mathcal{B}$ are independent of each other, we have $E[\Delta g \mid \mathcal{B}] = 0$

Using $g - \mathbb{E}[g] = (\bar{g} - \mathbb{E}[\bar{g}]) + \Delta g$ and expanding the squared norm, we obtain

$$\mathrm{Var}(g) = \mathrm{Var}(\bar{g}) + \mathbb{E}\|\Delta g\|_2^2 + 2\,\mathbb{E}\,\langle \bar{g} - \mathbb{E}[\bar{g}], \Delta g \rangle. \tag{17}$$

Consider the independence of cross-terms

$$\mathbb{E}\,\langle \bar{g} - \mathbb{E}[\bar{g}], \Delta g \rangle = \mathbb{E}_{\mathcal{B}}\left[\langle \bar{g} - \mathbb{E}[\bar{g}], \mathbb{E}[\Delta g \mid \mathcal{B}]\rangle\right] = 0. \tag{18}$$

Therefore,

$$\mathrm{Var}(g) = \mathrm{Var}(\bar{g}) + \mathbb{E}\|\Delta g\|_2^2, \tag{19}$$

i.e., the sampling-induced variance and the compression-induced (second-moment) term add up. Furthermore, the upper bound of the minibatch is a well-known conclusion, and we have

$$\text{Var}(\bar{\boldsymbol{g}}) \leq \frac{H(N-B)}{BN} \tag{20}$$

, where $H$ is a constant depending on data.This equation follows directly from the standard finite-population variance formula under sampling without replacement (see standard sampling theory).

Conditioned on a fixed mini-batch $\mathcal{B}$, the compression randomness is independent across samples. Moreover, as shown in Theorem 3.2(a), the weight-gradient estimator of linear layers is unbiased, which implies $\mathbb{E}[\Delta \boldsymbol{g} \mid \mathcal{B}] = \boldsymbol{0}$. Therefore,next we will focus on $\mathbb{E}\|\Delta\boldsymbol{g}\|_2^2$.

**Step 2: Bounding the compression-induced term $\mathbb{E}\|\Delta\boldsymbol{g}\|_2^2$.** Rethick the definition of $g$, it is obvious that

$$\Delta g \;=\; \frac{1}{B}\big(\text{vec}(\Delta\boldsymbol{G}_W^1),\; \text{vec}(\Delta\boldsymbol{G}_W^2),\; \ldots\big). \tag{21}$$

Noted that the 2-norm of a vector and the F-norm of a matrix can be interconverted. the squared Euclidean norm of $\Delta g$ decomposes exactly as

$$\|\Delta g\|_2^2 = \frac{1}{B^2}\sum_l \|\Delta\boldsymbol{G}_W^l\|_F^2. \tag{22}$$

Since activation compression is applied only to linear operators, the perturbation of the weight gradient at the $l$-th linear operator can be written as

$$\Delta\boldsymbol{G}_W^l = (\boldsymbol{G}_Z^l)^\top \Delta\boldsymbol{X}^l, \tag{23}$$

where $\Delta\boldsymbol{X}^l$ denotes the activation reconstruction error matrix and $\boldsymbol{G}_Z^l$ is the corresponding incoming gradient.

Substituting this expression and taking expectation, we obtain

$$\mathbb{E}\|\Delta g\|_2^2 = \frac{1}{B^2}\sum_l \mathbb{E}\big\|(\boldsymbol{G}_Z^l)^\top \Delta\boldsymbol{X}^l\big\|_F^2 \tag{24}$$

This equality provides an exact characterization of the variance contribution induced by activation reconstruction errors at each linear operator. Therefore, we have provided an upper bound for the variance:

$$\frac{H(N-B)}{BN} + \frac{1}{B^2}\sum_l \mathbb{E}\big\|(\boldsymbol{G}_Z^l)^\top \Delta\boldsymbol{X}^l\big\|_F^2. \tag{25}$$

## B.4. Proof of Corollary 3.5

From Theorem 3.4, we obtain explicit *upper bounds* on the variance terms appearing in the main-text rates. $\sigma^2$ denote an upper bound on the stochastic gradient variance, and $\sigma_c^2$ denote the upper bound on the compressed gradient variance. Theorem 3.4 provides computable bounds for these quantities. For notational convenience, throughout this proof we denote these bounds by

$$V_D := \sigma^2, \qquad V_C := \sigma_c^2 - V_D, \tag{26}$$

where $V_D = \frac{H(N-B)}{BN}$ and $V_C = \mathbb{E}\|\Delta g\|_2^2 = \frac{1}{B^2}\sum_l \mathbb{E}\big\|(\boldsymbol{G}_Z^l)^\top \Delta\boldsymbol{X}^l\big\|_F^2$.

Ghadimi and Lan (Ghadimi & Lan, 2013) show that the expectation of the gradient satisfies the following equation:

$$\mathbb{E}\|\nabla f(x_\tau)\|_2^2 \;\leq\; \frac{2L\Delta}{T} + \frac{\sqrt{8\sigma^2 L\Delta}}{\sqrt{T}}, \tag{27}$$

where $L$ is the $L$-smoothness constant of $f$, $\Delta$ is initial optimality gap (also a constant), and $T$ is the total number of iterations.

By Cauchy–Schwarz inequality,

$$\mathbb{E}\|\nabla f(x_\tau)\|_2 \le \sqrt{\mathbb{E}\|\nabla f(x_\tau)\|_2^2} \le \sqrt{\frac{2L\Delta}{T}} + \frac{(8\sigma^2 L\Delta)^{1/4}}{T^{1/4}}. \tag{28}$$

Under linear activation compression, the stochastic gradient estimator remains unbiased; hence $\mathbb{E}\|g - \nabla f(x)\|_2^2 = \mathrm{Var}(g)$. Substituting the result of Theorem 3.4, we obtain the following bounds.[2]

For standard SGD,

$$\mathbb{E}\|\nabla f(\boldsymbol{x}_\tau)\|_2 \ \le\ \mathcal{U}_T^S \triangleq \frac{\sqrt{2L\Delta}}{\sqrt{T}} + \frac{(8V_D L\Delta)^{1/4}}{T^{1/4}} \tag{29}$$

For compressed SGD,

$$\mathbb{E}\|\nabla f(\boldsymbol{x}_\tau)\|_2 \ \le\ \mathcal{U}_T^C \triangleq \frac{\sqrt{2L\Delta}}{\sqrt{T}} + \frac{(8(V_D + V_C)L\Delta)^{1/4}}{T^{1/4}} \tag{30}$$

Also rewrite the two bounds from equation (29) and (30) as

$$\mathcal{U}_T^S = a + b, \qquad \mathcal{U}_T^C = a + c, \tag{31}$$

where

$$a := \frac{\sqrt{2L\Delta}}{\sqrt{T}}, \quad b := \frac{(8V_D L\Delta)^{1/4}}{T^{1/4}}, \quad c := \frac{(8(V_D + V_C)L\Delta)^{1/4}}{T^{1/4}}. \tag{32}$$

Since $V_C \ge 0$, we have $c \ge b$. For $a, b > 0$, this implies

$$\frac{\mathcal{U}_T^C}{\mathcal{U}_T^S} = \frac{a + c}{a + b} \le \frac{c}{b} = \left(\frac{V_D + V_C}{V_D}\right)^{1/4} = \left(1 + \frac{V_C}{V_D}\right)^{1/4}. \tag{33}$$

Using the elementary inequality $(1 + x)^{1/4} \le 1 + x^{1/4}$ for all $x \ge 0$, we further obtain

$$\frac{\mathcal{U}_T^C}{\mathcal{U}_T^S} \le 1 + \left(\frac{V_C}{V_D}\right)^{1/4}. \tag{34}$$

Next, recall that

$$V_C = \frac{1}{B^2} \sum_l \mathbb{E}\|(\boldsymbol{G}_Z^l)^\top \Delta \boldsymbol{X}^l\|_F^2. \tag{35}$$

We have

$$V_C \le \frac{1}{B^2} \sum_l \mathbb{E}\big[\|\Delta \boldsymbol{X}^l\|_F^2 \|\boldsymbol{G}_Z^l\|_F^2\big]. \tag{36}$$

Under the assumption that the gradient is bounded , there exists a constant $G > 0$ such that

$$\|\boldsymbol{G}_Z^l\|_F^2 \le G^2 \quad \text{for all } l, \tag{37}$$

we obtain

$$V_C \le \frac{G^2}{B^2} \sum_l \mathbb{E}\|\Delta \boldsymbol{X}^l\|_F^2. \tag{38}$$

Combining the above inequalities yields

$$\frac{\mathcal{U}_T^C}{\mathcal{U}_T^S} \le 1 + \left(\frac{G^2}{B^2 V_D} \sum_l \mathbb{E}\|\Delta \boldsymbol{X}^l\|_F^2\right)^{1/4}, \tag{39}$$

---

[2]By absorbing constant factors and the (constant) optimality gap $\Delta$ into the $\mathcal{O}(\cdot)$ notation, (29)–(30) recover the $\mathcal{O}(\cdot)$-form bounds stated in the main text.

Absorbing the batch-size–dependent and data-dependent constants into the $\mathcal{O}(\cdot)$ notation, therefore

$$\frac{\mathcal{U}_T^C}{\mathcal{U}_T^S} \leq 1 + \mathcal{O}\left(\left(\sum_l \mathbb{E}\|\Delta \boldsymbol{X}^l\|_F^2\right)^{1/4}\right). \tag{40}$$

