# OpenReview forum: "Activation Compression in LLMs: Theoretical Analysis and Efficient Algorithm"
_ICML.cc/2026/Conference — Submitted to ICML 2026_

### Official Review · Reviewer_KddC · 2026-03-08

**Soundness:** 3
**Presentation:** 3
**Significance:** 2
**Originality:** 2
**Overall Recommendation:** 4
**Confidence:** 4

**Summary:**

The paper discusses which methods for compressing stored activations for the backward pass are safe and which are not.
It also proposes a method for compressing calculated gradients.

**Compliance With Llm Reviewing Policy:**

Affirmed.

**Final Justification:**

Added (mainly timing) experiments convinced me to raise my score to weak accept.

**Key Questions For Authors:**

- Did you really do all experiments in bfloat16 without mantissa correction?

**Limitations:**

Yes.

**Strengths And Weaknesses:**

The paper nicely discusses when one can use stored activation compression and proposes a good method for compressing calculated gradients. It is also well written and pleasant to read.
Experimental results show that the proposed method works, albeit with some loss in quality (see Figure 7 and Table 2).

What I do not like is the fact that one can get a better memory compression with clever engineering.
Gradient checkpointing is better than the proposed technique. One can also get rid of storing computed gradients and just directly apply them during the backward pass with a simple hook (this unfortunately makes the proposed gradient compression trick kind of pointless)
.
Authors also write: "All fine-tuning experiments are conducted in bfloat16". This is dangerous. Bfloat16 has 7bit precision. Let's say the weight update is 1000-times smaller than the weight (very typical case). Then the update would not be applied at all. And I do not see any tricks with extra mantissa like https://github.com/imoneoi/bf16_fused_adam. This lowers my trust in the experiment results.

---

> ### Author Rebuttal · Authors · 2026-03-31
>
> We thank the reviewer for raising important points.
>
> ### **Weakness 1: Limited Practical Advantage over Existing Engineering-Based Memory-Saving Approaches**
>
> We agree that engineering approaches such as gradient checkpointing and post-accumulate-grad hooks are  practical approaches.
> We discuss checkpoint in our paper and also considered hooks like register_post_accumulate_grad_hook.
> Nevertheless, we still believe that our proposed method remains valuable and complementary.
> Below, we clarify this from three aspects: Computational/time overhead, GPU memory reduction effectiveness, Compatibility.
>
> **Computational/time overhead:**
>
> Gradient checkpointing recomputes the forward pass, incurring significantly higher computational overhead.
> The hooks shorten gradient lifetime by performing per-layer updates during backward.
> However, they may reduce optimizer cross-parameter parallelism, leading to more fragmented GPU r/w and state updates, which result in higher computational overhead.
> Below, we report the runtime comparison of our method against  checkpoint and hooks on Llama3-3B trained on GSM8K:
>
> | Methods    | SFT   | Ours+ | Ours  | Hook  | Checkpoint |
> | ---------- | ----- | ----- | ----- | ----- | ---------- |
> | Time (min) | 56.75 | 47.76 | 62.88 | 63.48 | 86.49      |
>
> This highlights both the substantial overhead of engineering approaches and the advantage of our method.
> As model size and context length increase, the overhead of checkpoint becomes more pronounced.
>
>
> **GPU memory reduction effectiveness**
>
> Checkpoint and hooks reduce only activation memory and gradient memory, respectively, but do not address optimizer-state memory.
> In contrast, our proposed Ours+ enables simultaneous compression of activations, gradients, and optimizer states, resulting in greater overall GPU memory savings. We report the peak memory comparison on GSM8K using Llama3-3B (batch size 32) and Llama3-8B (batch size 4).
>
> **Llama3-8B:**
>
> | Methods    | Peak Allocated Memory (GB) |
> | ---------- | -------------------------- |
> | SFT        | 61.970                     |
> | Ours+      | 29.153                     |
> | Ours       | 54.980                     |
> | Checkpoint | 61.855                     |
> | Hook       | 56.534                     |
>
> **Llama3-3B:**
>
> | Methods    | Peak Allocated Memory (GB) |
> | ---------- | -------------------------- |
> | SFT        | 65.282                     |
> | Ours+      | 42.729                     |
> | Checkpoint | 44.661                     |
> | Ours       | 53.117                     |
> | Hook       | 63.342                     |
>
> The results demonstrate the advantage of our method in memory reduction.
> Moreover, these engineering approaches also lose their error-free advantage combined with optimizer compression[1], while our method show competitive accuracy.
>
> **Compatibility:**
>
> In multi-GPU training systems such as the ZeRO family[5], gradient synchronization and parameter gathering incur substantial communication and scheduling overhead, so a 'directly update during backward hook' is not a simple trick.
> Parameter updates in distributed training are not purely local: they must remain consistent with parameter/optimizer-state sharding. Immediate layer-wise updates would therefore require extra cross-device coordination. By contrast, our method compresses gradients at the representation level via low-rank structure, which is in principle more amenable to distributed training and may also reduce communication pressure.
>
> All in all, we believe that our work and this research direction remain valuable.
> This explain that despite the long-standing use of checkpointing, many current works[3,4] still focus on activation compression.
> One of the our contributions is to provide a theoretical foundation and guidance for activation compression in LLMs.
>
> ### **Weakness 2:Concerns About BF16 Numerical Precision**
>
> Representative works on optimizer compression, such as GaLore and APOLLO [1,2], use BF16 optimizer states to reduce optimizer-state memory. Therefore, we also use BF16 precision in our experiments to ensure a fair comparison.
> We additionally include results on Llama3-3B with GSM8K using an FP32 optimizer, and the difference from the BF16 results reported in the paper is little.
>
> | Methods | Score  |
> | :------ | :----- |
> | SFT     | 0.4586 |
> | Ours    | 0.4116 |
> | Ours+   | 0.4018 |
>
> We appreciate your helpful suggestion and incorporate your proposed tricks in future revisions.
>
> **Reference:**
>
> [1] GaLore: Memory-Efficient LLM Training by Gradient Low-Rank Projection. ICML 2024.
>
> [2] APOLLO: SGD-like Memory, AdamW-level Performance. MLSys 2025.
>
> [3] Memory-Efficient Fine-Tuning via Low-Rank Activation Compression. arXiv 2025.
>
> [4] HyC-LoRA: Memory Efficient LoRA Fine-tuning with Hybrid Activation Compression. MLSys 2025.
>
> [5] Zero-infinity: Breaking the gpu memory wall for extreme scale deep learning[C]. SC 2021.

---

> > ### Author Rebuttal · Reviewer_KddC · 2026-04-01
> >
> > Your answer about time savings (compared to grad. checkpoint) convinced me that this is a valuable avenue of work.
> >
> > But, I am confused about the answer to weakness 2.
> > Even nanogpt-speedrun uses an additional mantissa buffer for bf16 training (see https://github.com/KellerJordan/modded-nanogpt/blob/master/train_gpt.py#L568).
> >
> > And I do not understand the table in the answer. What part is bf16, what part is fp32?

---

> > > ### Author Response · Authors · 2026-04-07
> > >
> > > Thanks for your positive update and for recognizing the value of this direction.
> > > We are grateful for the opportunity to further clarify the remaining points.
> > >
> > > ### **Clarification on the Mantissa Buffer**
> > >
> > > Thank you for raising this important point.
> > > We would like to clarify that the experiments in our paper, as well as those in representative prior works on optimizer compression such as GaLore and APOLLO, are conducted under a BF16 optimizer setting without mantissa correction, in order to further minimize optimizer-state memory.
> > > We also carefully look into the mantissa buffer used in BF16 optimization and agree that it is a clever and practical technique for addressing the issue that some parameter updates may otherwise be lost.
> > > In the revised version, we will discuss and cite the mantissa buffer, and highlight this trade-off as an important limitation of current memory-efficient optimizer design.
> > >
> > > ### **Clarification on the Table in Our Previous Rebuttal**
> > > We clarify that the table in our previous rebuttal under **Weakness 2** reports results with an FP32 optimizer.
> > > The BF16 results are reported in the main paper.
> > > To make this clearer, below we  provide results, an additional BF16 setting with a mantissa buffer.
> > > Results are evaluated on Llama3-3B on GSM8K, with batch size 16, rank 8, and 10 training epochs.
> > >
> > > | Methods | BF16  | BF16 (mantissa buffer) | FP32  |
> > > |---------|-------|------------------------|-------|
> > > | SFT     | 0.466 | 0.453                  | 0.459 |
> > > | Ours    | 0.417 | 0.410                  | 0.411 |
> > > | Ours+   | 0.398 | 0.404                  | 0.401 |
> > >
> > > The slightly worse FP32 results in some cases are likely due to normal evaluation noise, since fine-tuning accuracy can only be measured at saved checkpoints rather than every update step. Nevertheless, the overall comparison suggests that the BF16 results are reliable, as the relative trends remain consistent across BF16 and FP32 settings.
> > >
> > > ---
> > > We sincerely thank the reviewer again for the valuable feedback and constructive suggestions.
> > > We hope that our responses adequately address your concerns.

---

### Official Review · Reviewer_1qje · 2026-03-10

**Soundness:** 4
**Presentation:** 4
**Significance:** 4
**Originality:** 3
**Overall Recommendation:** 5
**Confidence:** 3

**Summary:**

This paper establishes the theoretical guarantee of activation compression in linear layers and introduces a memory-efficient training approach by compressing activations and gradients using low-rank approximations.

**Compliance With Llm Reviewing Policy:**

Affirmed.

**Final Justification:**

This paper presents a theoretical analysis of the activation compression task, which I believe will be highly beneficial for future research.  On the empirical side, I initially raised concerns about the limited scale and limited evaluation metrics. The concern regarding the limited evaluation metrics was partially resolved, as the authors provided additional experiments showing consistent results. However, the concern regarding the limited scale remains. Although the additional experiments included results for an 8B model, considering the authors themselves acknowledged in the rebuttal that the benefits would be more apparent when using multiple GPUs on larger models, the absence of experiments on larger scales like 30B-70B is still a drawback. Nevertheless, despite this limitation in scale, while previous works have primarily designed activation compression methods empirically without theoretical justification, this paper is the first to provide a theoretical foundation for the activation compression task. For this reason, I believe the paper's contribution is sufficient for acceptance.

**Key Questions For Authors:**

1. The distinction between panels (a) and (b) in Figure 2 is not immediately clear. It would be helpful if the authors provided further clarification or a more detailed explanation of the differences between them.

**Limitations:**

yes

**Strengths And Weaknesses:**

**Strengths**

1. Unlike previous studies that depend on empirical heuristics, this paper provides a theoretical justification for the safety of activation compression in linear layers.

2. The proposed activation-gradient co-compression is an intuitive approach that can effectively lower training memory requirements.

3. There is a strong alignment between the theoretical claims and the experimental outcomes.

**Weaknesses**

1. The experiments are limited in scale, focusing only on small scale models like Llama3-3B and Qwen3-4B.

2. The high variance in performance across the datasets in Table 2 (GSM8K, MATH, GLUE) suggests that the method's impact on accuracy has high variance. To verify the robustness and generalization of the proposed method, more extensive evaluation on a more diverse set of benchmarks is necessary.

Although a more extensive evaluation involving a wider range of models and metrics would have strengthened the paper, the theoretical rigor and the provided experimental evidence constitute a sufficient contribution to the field. Therefore, I recommend the paper for acceptance.

---

> ### Author Rebuttal · Authors · 2026-03-31
>
> ### **Weakness 1: Limited Evaluation on Model Scale**
> Thanks for your suggeston.
> We further conduct a comprehensive set of experiments on Llama3-8B.
>
> We validate our theory through experiments on Llama-3.1-8B over 10 epochs on GSM8K, with rank 32 and batch size 4, by compressing different types of operators. The results provide further evidence for our theory.
>
> | Layer       | Score |
> |-------------|-------|
> | SFT         | 0.612 |
> | Attn_In     | 0.611 |
> | Attn_Out    | 0.620 |
> | MLP_In      | 0.603 |
> | MLP_Out     | 0.611 |
> | SiLU        | 0.135 |
> | RMSNorm     | 0.053 |
> | Softmax     | 0.015 |
> | All_Linear  | 0.589 |
>
> Under the same setting, we evaluated the accuracy of our proposed methods and other competitors.
> SFT serves as a no-compression baseline and provides a reference upper bound for accuracy score.
>
> **GSM8K:**
>
> | Method  | Score |
> |---------|-------|
> | SFT     | 0.612 |
> | Ours    | 0.581 |
> | LoRA    | 0.580 |
> | Ours+   | 0.552 |
> | Compact | 0.520 |
>
> **CommensenseQA:**
>
> | Methods | Score |
> |---------|-------|
> | SFT     | 0.822 |
> | Ours    | 0.824 |
> | LoRA    | 0.819 |
> | Ours+   | 0.812 |
> | Galore  | 0.799 |
> | CompAct | 0.778 |
>
> We also report the Peak Memory on dataset GSM8K.
> For the 8B model running on the A800, GPU memory limitations enforce a small batch size, making activation memory a less dominant component of the overall memory footprint.
>
> **Peak Memory:**
> | Methods    | Peak Allocated Memory (GB) |
> |------------|-----------------------------|
> | LoRA       | 23.630                      |
> | Ours+      | 29.153                      |
> | Galore     | 36.166                      |
> | Ours       | 54.980                      |
> | Checkpoint | 61.855                      |
> | SFT        | 61.970                      |
>
>
>
> ### **Weakness 2: Limited Benchmark Diversity**
> Thanks for your advice and we agree that evaluation on a more diverse set of benchmarks is necessary.
> We additionally conduct experiments on **CommonsenseQA** using Llama3.2-3B with batch size 8 and low-rank compression rank 8, further demonstrating our theory and the effectiveness of the proposed method.
>
> | Methods | Score |
> |---------|-------|
> | SFT     | 0.787 |
> | Ours    | 0.784 |
> | Ours+   | 0.769 |
> | Galore  | 0.766 |
> | LoRA    | 0.764 |
> | CompAct | 0.704 |
>
> Moreover, we would like to clarify that the performance variation across datasets is mainly due to the intrinsic difficulty differences among the benchmarks. This can be observed from the SFT baseline in Table 2, where the accuracies vary noticeably across GSM8K, MATH, and GLUE.
> In our view, the performance gap between our method and the SFT baseline is a more appropriate indicator, which shows that our method remains relatively stable across datasets.
>
>
> ### **Question 1: Detailed Explanation of the Difference Between Figure 2(a) and Figure 2(b)**
> Thanks for your helpful comment.
> We use a concrete example to clarify the difference between our proposed method (Fig. 2(b)) and the classical activation compression method (Fig. 2(a)).
> Consider a linear operator:  $\mathbf{Z} = \mathbf{X}\mathbf{W}^{\mathsf{T}}, \quad \mathbf{Z}\in\mathbb{R}^{m\times n},\ \mathbf{X}\in\mathbb{R}^{m\times d}$,
> whose weight gradient is
> $G_W = (\nabla_Z \mathcal{L})^T X$, which has shape $n\times d$.
> Figure 2(a) illustrates the workflow of the classical activation compression method.
> After the forward pass, the activation $X_{m\times d}$ is compressed and stored in the form of two low-rank matrices, $U_{m\times k}$ and $V_{d\times k}$.
> During backpropagation, the activation is first reconstructed as $\hat{X}_{m \times d} = U V^{\top}$,
> and the weight gradient is then computed as $\hat{G}_W=(\nabla_Z \mathcal{L})^T \hat{X}$, which has shape $n\times d$.
>
>
> Figure 2(b) corresponds to our proposed method, whose difference lies in the backward pass.
> Specifically, since
> $
> \hat{G}_W
> = (\nabla_Z L)^\top \hat{X}
> = \left( \frac{\partial \mathcal{L}}{\partial Z} \right)^\top (U V^\top)
> = \left( \left( \frac{\partial \mathcal{L}}{\partial Z} \right)^\top U \right) V^\top,
> $ $\hat{G}_W$ naturally admits a low-rank factorization.
> Therefore, our method does not first reconstruct $\hat{X}$. Instead, we directly stores $\left( \frac{\partial \mathcal{L}}{\partial Z} \right)^\top U$ and $\mathbf V^{\top}$
> as two low-rank factors of $\hat{G}_W$, and passes them to the optimizer. In this way, our method can  further reduce the memory footprint of gradient.

---

> > ### Author Rebuttal · Reviewer_1qje · 2026-04-01
> >
> > Thank you for the detailed rebuttal and for providing the additional experiments. I appreciate the effort to address the concerns regarding larger-scale evaluation, benchmark diversity, and the clarification of Figure 2. Overall, the rebuttal is helpful and strengthens the paper.
> >
> > **Weakness 1: Limited Evaluation on Model Scale**
> >
> > Thank you for adding the Llama-3.1-8B experiments. I think these additional results are useful, and including them in the final paper would make the empirical section stronger.
> >
> > That said, I still view this as only a partial response to the scalability concern. In the main paper, the 3B-scale evaluation is relatively thorough: it reports performance on GSM8K, MATH, and GLUE, as well as training-time comparisons across methods and memory analyses across methods and batch sizes. By contrast, the newly added 8B results mainly report task performance on GSM8K and CommonsenseQA and peak allocated memory. They do not yet provide training-time comparisons against various methods or memory-consumption results across different batch sizes. I understand that this is likely due to the limited time available during the rebuttal period, but the additional 8B experiments are still less comprehensive than the main experimental section.
> >
> > The 8B results also raise an additional practical question. In the rebuttal, you note that the 8B model use a small batch size on due to GPU memory limiations, which makes activation memory a less dominant part of the total memory footprint. In that setting, however, the method still shows higher peak memory than LoRA. This makes me wonder whether the practical advantage of the method may depend quite strongly on the training regime. Given current LLM standards, 8B is only a moderate scale, so I remain curious about how the method would behave on substantially larger models, such as 30B-70B. Without evidence at that scale, I still have some concern that the practical usefulness of the method may become more limited as the model size increases. Even if such experiments are beyond the current scope, I think this limitation should be discussed more explicitly in the final version.
> >
> > **Weakness 2: Limited Benchmark Diversity**
> >
> > Thank you for adding the CommonsenseQA experiment. I think this is a meaningful addition and helps make the empirical evaluation more convincing.
> >
> > While broader coverage across more benchmarks would still further strengthen the paper, the new CommonsenseQA result is a useful supplement to the original evaluation, and I appreciate that the authors took the time to add it during the rebuttal period.
> >
> > **Question 1: Detailed Explanation of the Difference Between Figure 2(a) and Figure 2(b)**
> >
> > Thank you for the additional explanation. The distinction between Figure 2(a) and Figure 2(b) is much clearer after reading the rebuttal.
> >
> > For the final version, I encourage the authors to revise Figure 2 and its caption so that readers can understand the difference more directly from the figure itself.
> >
> > Overall, I found the rebuttal helpful, and my overall assessment remains positive. The added CommonsenseQA and Llama-3.1-8B results strengthen the submission, and the clarification of Figure 2 resolves my earlier confusion. At the same time, my concern about practical scalability is only partially addressed, so I encourage the authors to incorporate the new results and to discuss the remaining limitations more clearly in the final version.

---

> > > ### Author Response · Authors · 2026-04-07
> > >
> > > We sincerely thank the reviewer for the positive assessment of our work and rebuttal, and for the constructive follow-up suggestions.
> > > We are grateful for the opportunity to further clarify the remaining concerns.
> > >
> > > ### **Limited Evaluation on Model Scale:**
> > >
> > > 1.
> > > Thanks for your constructive suggestions and positive feedback, which help improve the quality of our work.
> > >    We will supplement and incorporate these additional results into the revised version.
> > >    We apologize that, for the 8B model, the memory capacity of the A800 GPU prevented us from running experiments with larger batch sizes; therefore, we cannot provide memory results under different global batch sizes for the 8B setting.
> > >    In addition, we further supplement training-time comparisons on Llama-3-8B for CommonsenseQA and GSM8K (batch size = 4, rank = 32, epoch = 5), and report the results below.
> > >
> > > CommensenseQA:
> > >
> > > | Methods    | Time (min) |
> > > |------------|------------|
> > > | ours+      | 39.76      |
> > > | compact    | 40.09      |
> > > | baseline   | 56.49      |
> > > | checkpoint | 63.80      |
> > > | ours       | 74.76      |
> > > | galore     | 325.86     |
> > >
> > >
> > > GSM8K:
> > >
> > > | Methods    | Time (min) |
> > > |------------|------------|
> > > | ours+      | 44.70      |
> > > | compact    | 47.70      |
> > > | baseline   | 69.27      |
> > > | ours       | 74.55      |
> > > | checkpoint | 77.60      |
> > >
> > >
> > > 2.
> > > Thanks for raising this practical concern.
> > > The key distinction is that our method reduces activation memory, while LoRA does not.
> > > Therefore, the more activation memory contributes to the total memory usage, the greater the advantage of our method over LoRA.
> > > In our current 8B setting, we were limited to a small batch size due to hardware constraints, which reduces the contribution of activation memory. This explains why the memory advantage over LoRA is less pronounced in this case.
> > >
> > > In practice, in order to support larger models(30B/70B), longer-context tasks, and larger batch sizes,
> > > training  is distributed across multiple GPUs with model parallelism.
> > > Under such regimes, activation memory is expected to become a more important factor again.
> > > Therefore, our method can provide greater practical benefits and show a clearer advantage over LoRA than what is observed in the current 8B experiment.
> > > We acknowledge that experiments at 30B-70B scale are beyond the scope of the current submission and
> > > we will clarify this limitation in the revised version.
> > >
> > > ### **Weakness 2 & Question 1:**
> > > Thank you for your positive feedback and helpful suggestion, which help improve our work.
> > > We will improve our work accordingly in the revised version.
> > >
> > > ---
> > > We sincerely thank the reviewer for the overall positive assessment of our work and rebuttal, as well as for the constructive suggestions.
> > > We will incorporate the new results and discuss the remaining limitations in the revised version.

---

### Official Review · Reviewer_eWnk · 2026-03-11

**Soundness:** 3
**Presentation:** 3
**Significance:** 2
**Originality:** 2
**Overall Recommendation:** 4
**Confidence:** 4

**Summary:**

This paper investigates activation compression while training large language models. The authors show that, provided that an activation compressor is unbiased, then compressing input activations yields unbiased estimators of the weight gradients in linear operators. Furthermore, since activation compression does not affect the input gradients of linear operators, they derive convergence guarantees for standard SGD in which the convergence rate is perturbed by an additional term that depends on the activation reconstruction error. Building on this analysis, the authors restrict activation compression to the input activations of linear layers and represent them using low‑rank decompositions. This design allows the corresponding weight gradients to be expressed and updated directly in a low‑dimensional factorized form, enabling GaLore‑style optimizer updates without reconstructing full‑rank gradients. As a result, the method reduces not only activation memory but also the memory footprint of optimizer states, while additionally lowering the computational cost of gradient updates during training.

**Compliance With Llm Reviewing Policy:**

Affirmed.

**Final Justification:**

The authors have adequately addressed my main concerns. However, the novelty over LORACT seems to be limited.

**Key Questions For Authors:**

- You repeatedly emphasize that the theory is “LLM-tailored.” Concretely, which part of the framework relies on LLM/Transformer specifics versus applying to generic deep nets with linear/nonlinear operators?
- Could you also provide runtime measurements for Qwen3?
- Could you provide guidelines on how to select the rank for activation compression, either empirically or theoretically?
- Which activation compression method (RSVD vs RP) is used for each main experiment result?
- Is there a minor inconsistency in B.1. (b) (line 788), where it is stated that $D_{\Phi}[\Delta X]$ is linear, with this statement probably referring to $J_{\Phi}[\Delta X]$?

**Limitations:**

The authors do not clearly discuss limitations. See weaknesses for suggestions.

**Strengths And Weaknesses:**

**Strengths:**

1. The paper identifies a simple yet insightful fact that, for linear operators, unbiased activation reconstruction implies unbiased parameter gradients without propagating error into the input gradients, implying no upstream error propagation through compressed linear operators.
2. The authors show that low-rank activations yield low-rank gradients, enabling one to keep the optimizer states in a lower dimensional space. This enables further reducing memory requirements and compute cost.
3. Memory and compute cost reductions are verified empirically (Figure 1 and Table 3), supporting the claim that the approach can reduce memory footprint substantially.
4. For the most part, the paper is well written, and I appreciated the theoretical grounding for the approach.
5. The authors clearly articulate the challenges they are trying to solve.

**Weaknesses:**

1. Novelty may be incremental relative to prior low-rank activation compression + low-rank optimizer-state work. Low-rank activation compression has prior art (e.g., LORACT [1]) and low-rank optimizer state updates exist (e.g., GaLore); the most distinct contribution is the operator-level theory motivating “compress only linear ops” to control bias/propagation. The paper should more sharply delineate what is new beyond (i) “compress linear only” being common practice and (ii) existing low-rank training baselines.
2. “Ours” and “Ours+” are not good long-term identifiers. It would be better to assign a descriptive name to the algorithm variants.
3. The experimental results lack clarity in several places. In Figure 3, the training loss plots use different y‑axis scales across subplots, making the baseline (left) and activation‑compressed (center) trajectories appear more similar than they actually are. This makes it difficult to substantiate the claim that the method “only introduces minor perturbations to the optimization trajectory” (lines 371-372, column 1). In Figure 5, the main text indicates this figure reports Ours+ across batch sizes, but the caption/legend does not make this fully self-contained. Finally, although the paper mentions two activation‑compression methods (RSVD and random projection), it is not specified which method is used for the results reported in the main experiments.
4. There is limited scaling evidence across model sizes. Fine-tuning evaluates only Llama3 3B and Qwen3-4B, and pretraining uses Llama3 1B. It is hard to infer how memory savings and accuracy tradeoffs behave as model size grows (or as context length grows).
5. The paper discusses Ours vs Ours+ and shows some training-time and memory results, but does not provide a loss curve figure for Ours+. Given that Ours+ changes optimizer state handling, trajectory plots would strengthen the “minor perturbation” claim.
6. Table 1 is described as “training top−1 accuracy across epochs," but the SFT row decreases over epochs, which is unusual for training accuracy and suggests either (i) the metric is actually evaluation accuracy or (ii) the labeling needs clarification. Additionally, Table 2 reports task accuracy for different methods but does not show a “base (pre-finetune) model” score; including it could help contextualize absolute gains from fine-tuning vs compression choices.
7. The rank selection guidance is underdeveloped. The paper evaluates ranks 8 and 32 and reports reconstruction error stats, but it does not provide actionable guidance for choosing rank given a memory/accuracy budget (beyond observing training loss/accuracy).
8. The main text focuses exclusively on supervised fine‑tuning. Pretraining results are relegated to Appendix A.3, where GaLore performs strongly; the main paper would benefit from a clearer discussion of when activation compression helps relative to optimizer-state-only methods in pretraining regimes where activations dominate memory, and why the relative behavior differs from the fine-tuning setting in Table 2.
9. Definitions and notation appear as the argumentation progresses, making the theoretical discussion difficult to follow throughout the paper. Given the breadth of notation, a consolidated section would be helpful.
10. The paper relies on the assumption that $\hat{X}$ is an unbiased estimator of $X$ (i.e., $\mathbb{E}[\Delta X] = 0$), but there is no discussion on whether the compression methods used (RSVD and random projection) satisfy this assumption.
11.  The readability of Theorem 3.4. would be improved by explicitly stating which expectations/randomness are over the mini-batch draw versus compression randomness, and how the two sources are separated/assumed independent.
12. There are minor grammatical errors, e.g. "derive gradient variance bound" instead of "derive a gradient variance bound" (line 23, column 1) and "wery" instead of "very" (line 316, column 2).
13. The convergence discussion follows classical SGD theory (starting at line 258, column 1), yet the experimental setup uses AdamW (lines 291-292, column 2). Even a short discussion of how/why the theory is expected to transfer (or what does not transfer) would make the claims more precise.

[1] Shi et al. "Memory-Efficient Fine-Tuning via Low-Rank Activation Compression." arXiv (2025).

---

> ### Author Rebuttal · Authors · 2026-03-31
>
> Thank you for all the valuable suggestions. We address your concerns as follows.
>
> **Weakness 1:**
>
> Optimizer-state compression methods such as GaLore mainly reduce optimizer memory, while activation compression methods such as LORACT mainly reduce activation memory. (We are very familiar with LORACT.)
> Besides theory, our contribution is an operator-level view that enables joint compression of activations and optimizer states by using gradients as the bridge between them without substantial extra compression overhead, maintaining competitive runtime and accuracy.
>
> **Weakness 2:**
>
> We will rename our method as AG-CC (Activation–Gradient Co-Compression) and AGO-CC (Activation–Gradient-Optimizer Co-Compression) in
> the revision.
>
> **Weakness 3 & 5 & Question 4:**
>
> We will revise the figure/caption and add the loss curve for Ours+ in the next version.
> Ours uses RSVD, while Ours+ uses RP.
> To support the “minor perturbation” claim, we add the following loss values:
>
> | Epoch | Baseline | Ours(rank=32)  | Ours+ (rank=32) |
> |--------------|----------|----------------|-----------------|
> | 2            | 0.296    | 0.375          | 0.409           |
> | 4            | 0.161    | 0.245          | 0.330           |
> | 6            | 0.100    | 0.207          | 0.298           |
> | 8            | 0.076    | 0.194          | 0.285           |
> | 10           | 0.069    | 0.190          | 0.281           |
>
> **Weakness 4:**
> We conduct comprehensive experiments on Llama3-8B in our response to **Reviewer #3, Weakness 1**.
>
> **Weakness 6:**
> We apologize for the typo. The metric reported here is evaluation accuracy, not training accuracy.
> The pre-finetune scores are 0/0/0.432 for LLaMA3-3B on GSM8K/MATH/GLUE and 0.628/0.104/0.432 for Qwen3-4B.
> Performance depends on both fine-tuning and compression strategy.
>
> **Weakness 7 & Question 3:**
>
> Previous activation compression for LLM works generally do not provide a principled rank-selection strategy.
> Our theory suggests choosing the rank, under a memory budget, to minimize the extra gradient variance from activation compression,
> i.e., the term in Theorem 3.4, $\mathbb{E}|(G_Z)^\top \Delta X|_F^2$.
> In practice, we estimate this quantity on a small calibration set before training.
> Since rank 8 already works well, we will further explore this in subsequent research.
>
> **Weakness 8:**
>
> 1.The main difference is that in pretraining, much longer contexts make activations far larger than optimizer states, so activation compression introduces larger approximation error; this helps explain why GaLore performs better there.
>
> 2.The advantage of activation compression is that it directly targets the dominant source of memory usage.
> Compressing only the optimizer states is far from sufficient for memory efficiency.
>
> **Weakness 9 & 12 & Question 5:**
>
> We will improve the organization of notation and fix the minor writing errors in the revision.
>
> **Weakness 10:**
>
> Thanks for raising this important point.
> We address this issue in our response to **Reviewer #1, Weakness 2**, and would be very grateful if you could take a look.
>
> **Weakness 11:**
>
> 1. The first term is induced by mini-batch sampling, and the second is induced by activation compression.(We clarify it in our paper.)
> 2. The proof does not require $\Delta g$ to be independent of the mini-batch; it only uses that the compression randomness is independent of mini-batch sampling and that  for a fixed mini-batch, the activations are fixed, so Theorem 3.2(a) gives $\mathbb{E}[\Delta g \mid \mathcal{B}]=0$ for linear layers.
>
>
> **Weakness 13:**
>
> SGD-style stochastic gradient analysis provides a useful foundation for understanding AdamW, and our gradient analysis and variance bound can transfer to AdamW. Because convergence guarantees for AdamW are delicate and generally require additional assumptions, prior work typically analyze convergence under SGD.
>
> **Question 1:**
> Our theory is not limited to LLMs and can also apply to generic deep networks.
> We call it “LLM-tailored” mainly because:
>
> (1)Our analysis is especially relevant for very deep models, where compression errors across nonlinear operators accumulate more significantly.
>
> (2)While the theory is general, our practical activation-gradient joint compression scheme is instantiated on linear operators. Since most gradients, and optimizer states in Transformers are concentrated in linear layers, the method is especially well suited to LLMs.
>
> (3)The GPU memory issue is mainly pronounced in LLMs.
>
> **Question 2:**
> Below we provide the training time and  throughput(token/second) using Qwen3-4B on GSM8K.
>
> | Methods     | Ours+  | LoRA   | SFT     | Ours   | Galore |
> |------------|--------|--------|---------|--------|--------|
> | Throughput | 5948.84| 5190.54| 5006.26 | 4576.03| 2499.84 |
>
> | Methods    | Ours+ | LoRA  | SFT   | Ours  | Galore |
> |------------|-------|-------|-------|-------|--------|
> | Time (min) | 76.80 | 88.02 | 91.26 | 99.84 | 182.76 |

---

> > ### Author Rebuttal · Reviewer_eWnk · 2026-04-02
> >
> > Thank you for addressing my main concerns. The novelty over LORACT seems to be limited, and there is a gap between theory (SGD) and experimentation (AdamW) that is not well explained (although the argumentation is acceptable). I have raised my score, which I believe is a fair assessment.

---

> > > ### Author Response · Authors · 2026-04-07
> > >
> > > We sincerely thank the reviewer for the constructive feedback and updated assessment.
> > > We are grateful for the opportunity to further clarify the remaining points.
> > >
> > > ### **Novelty relative to LORACT:**
> > > We thank the reviewer for raising this important point.
> > > We agree that our work is related to LORACT in that both study low-rank activation compression.
> > > However, we believe that our contribution is meaningfully distinct in three aspects:
> > > (1) the compression target, (2) the fine-tuning regime, and (3) the scope of memory reduction.
> > > (1) LORACT compresses activations at RMSNorm, whereas our theory suggests that compressing activations at nonlinear operators can introduce instability; accordingly, we compress activations at linear operators, which is directly supported by our analysis and leads to better accuracy.
> > > (2) Our method is designed for the SFT regime, aiming to remain as close as possible to full SFT, while LORACT is studied in the LoRA-based PEFT regime and is therefore tied to the LoRA baseline.
> > > (3) Our Activation–Gradient Co-Compression framework reduces not only activation memory, but also gradient and optimizer-state memory, which we view as a substantive distinction.
> > >
> > >
> > > Empirically, we additionally compare the accuracy performance of our method against LORACT on Llama3-3B across GSM8K, GLUE, and CommonsenseQA, as shown below.
> > >
> > > | Methods | GSM8K | GLUE | CommonsenseQA |
> > > |-------------------|-------|------|----------------|
> > > | SFT               | 0.466 | 0.840 | 0.787 |
> > > | Ours              | 0.417 | 0.771 | 0.784 |
> > > | Ours+             | 0.398 | 0.800 | 0.769 |
> > > | LORACT            | 0.014 | 0.716 | 0.762 |
> > >
> > > Across all three benchmarks, our method outperform LORACT.
> > > LORACT appears less robust in chain-of-thought setting, while the gap is smaller but still consistent on the classification-oriented benchmarks.
> > > Overall, compared with LORACT, our method is theoretically motivated in selecting the compression target and extends compression beyond activations to gradients and optimizer states via Activation-Gradient Co-Compression rather than relying on LoRA. These differences lead to stronger empirical performance in our setting.
> > > We will add a brief discussion of this connection in the revised version.
> > >
> > > ### **Gap Between SGD Theory and AdamW Practice**
> > > We thank the reviewer for this helpful suggestion. We agree that a brief discussion of the relationship between our SGD-based theory and AdamW-based experiments would improve clarity.
> > >
> > > Our theory is intended to analyze the activation compression mechanism itself, rather than a specific optimizer, which are broadly informative.
> > > In particular, it establishes gradient unbiasedness and a variance bound under activation compression.
> > > We present convergence under SGD because proving convergence for AdamW requires additional assumptions[1],
> > > and the SGD setting provides a more transparent view.
> > >
> > > At the same time, our analysis can serve as a building block for studying AdamW. Prior work has shown that, under additional conditions on the parameters and initialization, together with the standard $L$-smoothness assumption, AdamW convergence can be established from unbiased gradient estimates with bounded variance [2]. Since our analysis already provides these properties under activation compression, it is compatible with such an extension. We will add a brief discussion of this connection in the revised version.
> > >
> > > ### **Reference:**
> > > [1] The Curious Case of AdamW[EB/OL]. OpenReview, 2025.
> > >
> > > [2] On the $O(\frac{\sqrt d}{K^{1/4}})$ Convergence Rate of AdamW Measured by $ℓ_1$ Norm. NeurIPS, 2025.
> > >
> > > ---
> > > Thanks again for your valuable suggestions and positive feedback on our work.

---

### Official Review · Reviewer_EAnv · 2026-03-13

**Soundness:** 4
**Presentation:** 3
**Significance:** 3
**Originality:** 3
**Overall Recommendation:** 4
**Confidence:** 3

**Summary:**

This paper tackles the memory bottleneck from storing intermediate activations in LLM training. It clearly shows why unbiased compression works fine for linear operators but amplifies errors in nonlinear ones. Building on this, the authors introduce a really neat 'activation-gradient co-compression' approach. By reusing low-rank activation factors, they manage to compress weight gradients with basically zero extra compute. The empirical results on LLaMA3 and Qwen look solid, showing impressive memory savings over baselines like LoRA and GaLore while keeping accuracy competitive.

**Compliance With Llm Reviewing Policy:**

Affirmed.

**Final Justification:**

The rebuttal addressed my concerns and I raised my score.

**Key Questions For Authors:**

See weakness.

**Limitations:**

yes

**Strengths And Weaknesses:**

Strength:
1. I really appreciate the clear mathematical distinction between compressing linear vs. nonlinear operators. Proving that linear compression keeps gradients unbiased and prevents upstream error propagation is a great contribution—it puts rigorous math behind what many in the community already suspected.
2. The activation-gradient co-compression is simple but brilliant. Recognizing that $G_W=(\nabla_Z \mathcal{L})^T X$ and reusing the low-rank activation factors ($X=UV^T$) to keep the gradient factorized is a very neat trick. It cuts down memory without adding extra compute overhead.

Weakness:
1. As we know, LLM activations have massive, structured outliers. Since the method relies on RSVD and Random Projection, I'm a bit concerned that low-rank approaches usually struggle to capture sparse outliers, which could blow up the Frobenius norm error $||\Delta X||_F^2$. I'd love to see some discussion on how these outliers affect the unbiasedness assumption and variance bounds.
2. There's a critical gap between the theoretical framework and the empirical implementation that needs addressing. The core theory (especially Theorem 3.2 and the variance bounds in Theorem 3.4) strictly assumes unbiased activation compression, where $\mathbb{E}[\Delta X] = 0$. However, the experiments heavily rely on Randomized SVD (RSVD). As we know, SVD-based low-rank truncation is inherently biased because it systematically discards the orthogonal complement (the long tail of singular values). If the practical compression is biased, the theoretical guarantees seem to fall apart. I would really appreciate it if the authors could reconcile this fundamental contradiction—how do the theoretical bounds actually hold up when using biased RSVD in practice?
3. Using saved_tensors_hooks is just a proxy. It would be much stronger to report actual peak allocated memory and true end-to-end throughput (Tokens/sec) rather than just theoretical estimates.

---

> ### Author Rebuttal · Authors · 2026-03-31
>
> Thanks for all your valuable feedback on our paper. We address your concerns as follows.
>
> ### **Weakness 1: Impact of Sparse Outliers on Frobenius-Norm Reconstruction Error:**
>
> Outliers themselves do not invalidate the unbiasedness, and they do not necessarily harm low-rank decomposition methods.
> We will discuss this from both the experimental and theoretical perspectives.
>
> Empirically, we conduct the experiments on two datasets, GSM8K and the GLUE subset CoLA, using Llama3-3B.
> We randomly selected a fixed linear operator and examined the element-wise distribution of activations across 10 epochs.
> In both cases, the distributions are approximately Gaussian.
> For GSM8K, the mean  and variance are approximately $-0.001$ and $0.01$, respectively;
> for CoLA, they are approximately $0.002$ and $0.077$, respectively.
> We further analyzed the element-wise distribution of the reconstruction error $\Delta \mathbf{X}$ for RSVD.
> For GSM8K, all values lie within the range of $[-0.9, 0.6]$, and 99.8\% of them fall within $[-0.2, 0.2]$.
> For CoLA, all values lie within $[-0.48, 0.54]$, and 99.8\% of them fall within $[-0.15, 0.15]$.
> These results suggest that the low-rank approximation is robust to outliers, and their effect on the Frobenius-norm error is limited.
>
> Theoretically, prior classical work [1] has established rigorous upper bounds on $\mathbb{E}\lVert \Delta \mathbf{X}\rVert_F$ for both RSVD and RP; For both methods the bound is governed mainly by the singular-value distribution of $\mathbf{X}$.
> As a result, outliers may increase $\lVert \Delta \mathbf{X}\rVert_F$ indirectly, namely by modifying the spectral structure of $\mathbf{X}$,
> which is hard to judge in general.
>
>  Overall, the influence of outliers appears limited.
>
> ### **Reference**
> [1] Finding structure with randomness: Probabilistic algorithms for constructing approximate matrix decompositions. SIAM Review.
>
> ### **Weakness 2: Mismatch Between the Unbiased Compression Assumption and the Biased RSVD Implementation:**
>
> For RP, conditioned on a fixed $\mathbf{X}$, we have $\mathbb{E}[\Delta \mathbf{X}\mid \mathbf{X}] = \mathbf{0}$, which is fully consistent with our theory.
> RSVD does not satisfy $\mathbb{E}[\Delta \mathbf{X}\mid \mathbf{X}] = \mathbf{0}$. To better understand its practical behavior, we additionally evaluate its aggregate bias under activations, i.e., $\mathbb{E}[\Delta \mathbf{X}] \approx \mathbf{0}$, which is expectation over activations, rather than the fixed-input conditional expectation used in the clean theory. This helps substantially narrows the gap between theory and practice.
>
> To empirically assess the bias, we evaluate LLaMA-3.2-3B on GSM8K and GLUE subset CoLA.
> At each epoch, we randomly select a linear operator and record its activations over the dataset.
> We calculate the dataset-level expectation matrix $\mathbb{E}[\Delta \mathbf{X}]$, and use its mean absolute value $\frac{1}{mn}\|\mathrm{vec}(\mathbb{E}[\Delta \mathbf{X}])\|_{1}$ as a scalar measure of bias.
> The results are presented below:
>
> **GSM8K:**
>
> | Epoch | RSVD ($×10^{-3}$) | RP($×10^{-1}$) |
> | ----- | ----------------- | -------------- |
> | 1     | 2.531             | 0.821          |
> | 2     | 2.855             | 0.744          |
> | 3     | 2.959             | 0.763          |
> | 4     | 3.081             | 0.746          |
> | 5     | 3.044             | 0.738          |
>
> **COLA:**
>
> | Epoch | RSVD ($×10^{-3}$) | RP($×10^{-1}$) |
> | ----- | ----------------- | -------------- |
> | 1     | 5.548             | 14.31          |
> | 2     | 5.689             | 14.56          |
> | 3     | 5.820             | 14.46          |
> | 4     | 6.127             | 13.59          |
> | 5     | 6.147             | 13.31          |
>
> According to the results, the aggregate bias of RSVD is much smaller than that of RP.
> Theoretically, our finding in **Weakness 1** above that the activation distribution is approximately symmetric, providing the explanation for this phenomenon, and helps mitigate the concern about the bias of RSVD.
>
> ### **Weakness 3: Lack of End-to-End Evaluation:**
>
> In our paper, we reported peak GPU memory usage in Table 8  and runtime in Table 3.
> We additionally report the peak memory and throughput(token/second) on Llama-3.2-3B with GSM8K at rank 8, highlighting the advantage of our method.
>
> | Methods    | Peak Allocated Memory (GB) |
> | ---------- | -------------------------- |
> | Ours+      | 42.729                     |
> | Checkpoint | 44.661                     |
> | Ours       | 53.117                     |
> | Galore     | 54.822                     |
> | LoRA       | 56.764                     |
> | SFT        | 65.282                     |
>
> | Methods    | Ours+   | SFT     | LoRA    | Ours    | Checkpoint | Galore  |
> | ---------- | ------- | ------- | ------- | ------- | ---------- | ------- |
> | Throughput | 8790.01 | 7397.55 | 7166.45 | 6676.38 | 4853.87    | 3831.79 |

---

> > ### Author Rebuttal · Reviewer_EAnv · 2026-04-03
> >
> > Thank you for your rebuttal, my main concerns are well-addressed. Accordingly, I will raise my score to weak accept.

---

> > > ### Author Response · Authors · 2026-04-07
> > >
> > > We sincerely thank the reviewer for the recognition of our work and for the positive update.
> > > We are pleased that our responses have addressed your concerns and your constructive suggestions helped us strengthen the quality of our work.
> > > We sincerely appreciate your time, careful evaluation, and support for our work.

---

### Decision · Program_Chairs · 2026-04-30

**Decision:**

Reject

**Comment:**

The paper studies activation compression in LLMs to save memory during training. It is interesting that the paper argues a good activation compression requires unbiased and low-variance estimation, though these results are standard in traditional optimization. After going through the paper, I have concerns about the experiments. From my perspective, large activation compression should be used in pre-training, since memory and computation savings are less critical in fine-tuning. However, pre-training experiments are not included in the main text. I suggest the authors revise the paper and conduct more challenging experiments.